# Mitigating Forgetting in LLM Fine-Tuning via Low-Perplexity Token Learning

**Chao-Chung Wu**[1*]   **Zhi Rui Tam**[1*]   **Chieh-Yen Lin**[1]   **Yun-Nung Chen**[2]
**Shao-Hua Sun**[1,2†‡]   **Hung-yi Lee**[2†]
[1]Appier AI Research, [2]National Taiwan University
{johnson.wu, ray.tam}@appier.com

## Abstract

Maintaining consistent model performance across domains is a fundamental challenge in machine learning. While recent work has explored using LLM-generated data for fine-tuning, its impact on cross-domain generalization remains poorly understood. This paper presents a systematic analysis revealing that fine-tuning with LLM-generated data not only improves target task performance but also reduces non-target task degradation compared to fine-tuning with ground truth data. Through analyzing the data sequence in tasks of various domains, we demonstrate that this enhancement of non-target task robustness stems from the reduction of high perplexity tokens found in LLM-generated sequences. Following our findings, we showed that masking high perplexity tokens in ground truth training data achieves similar non-target task performance preservation, comparable to using LLM-generated data. Extensive experiments across different model families and scales, including Gemma 2 IT 2B, Llama 3 8B Instruct, and three additional models, agree with our findings. To the best of our knowledge, this is the first work to provide an empirical explanation based on token perplexity reduction to mitigate catastrophic forgetting in LLMs after fine-tuning, offering valuable insights for developing more robust fine-tuning strategies.

## 1   Introduction

Supervised fine-tuning large language models (LLMs) [29] has proven highly effective for enhancing their ability to follow novel instructions [15, 21, 23, 43, 49] and produce useful outputs across a wide range of tasks, including summarization and web querying [1, 17, 26, 34]. As LLMs are increasingly applied in specialized domains such as arithmetic and programming assistance [27, 38], practitioners often encounter substantial computational and data cleaning issues when adapting these models to gain improvements. These difficulties are exacerbated by performance saturation in smaller LLMs, where benchmark improvements on tasks like MATH [12] and MBPP [3] tend to plateau as model size decreases. Moreover, fine-tuning can risk degrading a model's general capabilities [4, 24, 30, 31]. This raises a critical question: how can we efficiently fine-tune pre-trained, instruction-following LLMs for domain-specific tasks *without compromising other skill sets such as arithmetic or common sense reasoning*, especially when the original fine-tuning data is unavailable or computational resources are constrained?

Recent work has shown that fine-tuning with model-generated data can be highly effective, often outperforming training on ground truth data when using high-quality generated samples [10, 32,

---

[*]Equal contribution

[†]Equal advisory contribution

[‡]Shao-Hua Sun is an Adjunct Research Scientist at Appier AI Research
  Our code and datasets are available at https://github.com/appier-research/robust-llm-finetunes

Figure 1: An example MATH problem showing more high perplexity tokens (highlighted in **red**, perplexity $\geq 2.5$) in ground truth than Self-Output responses (Llama 3 self-generated responses).

35, 48]. For example, Ren et al. [31] demonstrated that model-generated responses can surpass human annotations in downstream performance, particularly when using more capable models (*e.g.*, GPT-4) to generate training data and mitigate degradation on general tasks. Although they attributed part of this success to the "familiarity" of generated responses, measured via low perplexity, it remains unclear why such data works well for instruction tuning. Specifically: (1) The regenerated response may differ contextually from the ground truth, and such contextual bias could affect the utility of low-perplexity training. (2) The "non-forgetting" impact of low-perplexity training on the performance of tasks not being fine-tuned, which maintains the practicality of the model, is still underexplored. (3) Since model size plays a role in distillation benefits, it is important to disentangle the effects of model scale and evaluate low-perplexity training using models of the same size.

Our work offers an empirical explanation, grounded in loss and perplexity reduction, for why LLM-generated data tends to cause less degradation in fine-tuning outcomes. Through systematic analysis, we first show that fine-tuning with different types of LLM-generated data (*e.g.*, self-generated and rephrased responses) significantly mitigates degradation on non-target tasks compared to fine-tuning with ground truth data. Second, we trace this improvement to a key property illustrated in Figure 1: LLM-generated responses exhibit lower sentence-level perplexity and a smaller proportion of high-perplexity tokens. This leads to a practical insight: by simply masking high-perplexity tokens in ground truth training data, we can achieve a comparable reduction in non-target task degradation as observed with LLM-generated data. Importantly, this can be done using the same model prior to training, allowing us to pre-process the data and harness the benefits of low-perplexity training.

We propose a novel empirical explanation for non-target task performance degradation, introducing Selective Token Masking (STM), a simple yet effective strategy that filters out high-perplexity tokens to enable low-perplexity training. We have applied STM on (1) different perplexity criteria, (2) different model families and scales, and (3) various fine-tuning techniques from full-weight fine-tuning to parameter-efficient fine-tuning. We found that STM could achieve almost the same capabilities as the original models have on non-target tasks after fine-tuning on target tasks under all circumstances. We could identify that the degradation results from the low model weights update during the low-perplexity training. Together, these insights provide a new perspective on designing fine-tuning strategies that better retain general capabilities while enhancing performance on domain-specific tasks.

## 2 Training LLMs on self-generated data

To evaluate the effectiveness of LLM-generated data for fine-tuning, we explore two distinct methods for generating training data across two different target tasks. We then assess the models fine-tuned with the generated data on five non-target tasks, respectively. The two generation strategies, Self-Output and Rephrase, offer complementary approaches to constructing LLM-based training data, each addressing different trade-offs and challenges. In this section, we introduce the original datasets used for data generation and describe our methodology for constructing self-generated training datasets using language models.

## 2.1 Training and evaluation framework

For target-task fine-tuning, we adopt the MBPP and MATH datasets. These datasets provide rich annotations, including full solutions, reasoning steps, and test cases, beyond just the final answers, allowing models to learn comprehensive task-solving behaviors. For evaluation, in addition to using the test sets from MBPP and MATH, we assess model performance on GSM8K [7], ARC-Challenge [6], and BIRD [19] to examine generalization to non-target tasks involving various forms of reasoning and generation. A detailed description of the datasets is provided in Section 4.

## 2.2 Self-Output

Self-Output follows a similar high-level approach, similar to other synthetic data generation techniques [31, 48], where a high-quality synthetic dataset is created by sampling diverse responses from an LLM and applying a strict filtering process. For each training instance, we use a language model $M$ (*e.g.*, Llama 3 8B Instruct) to generate $N$ (*e.g.*, 32) distinct responses with a temperature setting of $T = 0.7$. The generated responses are then filtered to those that are semantically aligned with the ground truth, ensuring the quality and relevance of synthetic data.

Self-Output excels at producing high-quality training data in domains with reliably verifiable outputs (*e.g.*, programming tasks with objective ground truths).

Table 1: Perplexity average and variance over answer sequences of MBPP and MATH training datasets on Llama 3 8B Instruct. The average sentence perplexity is calculated by averaging the sum of each sentence's perplexity over the token perplexity score.

| Data | Method | Avg. PPL |
|------|--------|----------|
| MBPP | Ground Truth | 4.83 (7.04) |
|      | Rephrase | 1.69 (0.16) |
|      | Self-Output | 1.16 (0.01) |
| MATH | Ground Truth | 2.45 (0.81) |
|      | Rephrase | 2.11 (9.28) |
|      | Self-Output | 1.34 (0.03) |

However, it remains constrained to settings where such verification is feasible and imposes significant computational overhead from the multiple generations and filtering required to identify valid outputs.

## 2.3 Rephrase

Yang et al. [47] proposed self-distillation. The process uses an instruction-finetuned LLM to rephrase ground-truth responses in its style. By providing both the instruction and ground truth to the target LLM, it generates semantically equivalent reformulations with ground truth responses. Compared to Self-Output, Rephrase is computationally efficient, as it requires only a single pass of generation without additional filtering. However, Rephrase is susceptible to hallucinations, as evidenced by the higher variance in token-level perplexity (see Table 1). Despite this limitation, Rephrase offers versatility, as it does not rely on verifiable ground truth and can be applied to a broader range of tasks.

We have a sanity check on the output of Rephrase, and remove the incorrect label output as the final training set for Rephrase to ensure that all the final outputs of Rephrase are correct. But we did not evaluate the intermediate output of Rephrase, *e.g.*, the reasoning or CoT process. This process is partially reflected in Appendix F, Table 17.

Table 2 presents the performance comparison between Self-Output, Rephrase, and Baseline Fine-tuning on ground truth using different models in terms of model size and model series across both target task improvement and non-target task degradation rate. Both task improvement rate (TI) and degradation rate (BWT) are adopted from Backward Transfer rate from [14] and modified as follows:

$$\mathbf{TI} = (a_{target}^{(train)} - a_{target}^{(original)})/a_{target}^{(original)}. \tag{1}$$

$$\mathbf{BWT} = \frac{1}{T-1}\sum_{i=1}^{T-1}(a_i^{(train)} - a_i^{(original)})/a_i^{(original)}. \tag{2}$$

Given $T$ tasks composed of one target task and $T-1$ non-target tasks, **TI** is the target task performance improvement percentage after training on the target task. **BWT** refers to the average performance degradation percentage of the $T - 1$ non-target task after training on the target task. The models to fine-tune in this paper refer to generic instruction following models before fine-tuning with target task data, *e.g.*, Llama 3 8B Instruct [8], Gemma 2 IT 2B [36], Mistral 7B Instruct [16], and OLMo 2 7B series [40]. For target task performance, Self-Output demonstrates consistent positive **TI** in both MBPP and MATH. For the non-target task, the Self-Output method achieves better BWT scores

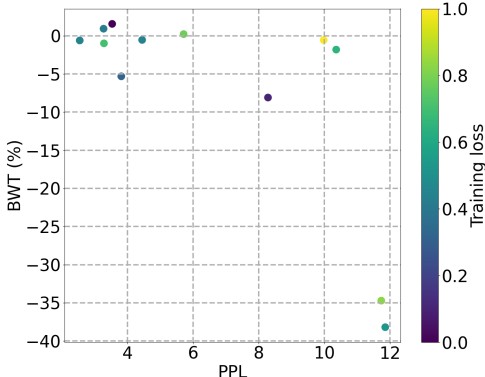
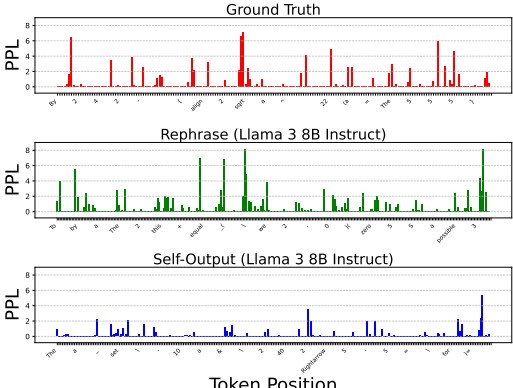

Figure 2: Fine-tuned Llama 3 8B Instruct on MBPP labels generated by various LLMs. Backward Transfer (BWT) measures performance drop on non-target tasks, showing that Low-perplexity labels training generally leads to less degradation (upper left) than ground truth (bottom right).

Figure 3: Comparison of token-level perplexity (PPL) distributions between human-annotated Ground Truth (top) and Llama 3 8B Instruct generations for Rephrase and Self-Output sequence (middle and bottom), where PPL of Self-Output data is low and with fewer spikes.

than the baseline fine-tuning in most cases, indicating much less degradation on non-target task capabilities. For more details, such as performance comparison with other models or other minor findings for self-output and rephrase method, please refer to Appendix D.

## 2.4 Analysis of perplexity across each sample

In this section, we analyze the perplexity of datasets generated by Self-Output and Rephrase to quantify their impact on model uncertainty and fine-tuning stability. We first collect the inference statistics of GPT-4o, GPT-4o mini, Gemma 2 IT 27B, Llama 3 8B Instruct, and Gemma 2 IT 2B from all three methods: Baseline Fine-tuning, Rephrase, and Self-Output strategy, along with their BWT, as Figure 2 shows that a higher perplexity of target training data could result in a higher degradation on non-target task. Second, we further collect the Llama 3 8B Instruct inference statistics from three kinds of training data: Baseline Finetuning (ground truth data), Rephrase, and Self-Output strategy. We evaluate the perplexity of responses by summing up the negative log probability of each token conditioned on its previous context in the sentence to measure the uncertainty of model outputs over the entire sequence. Specifically, the perplexity $PPL$[4] across a sequence is computed as:

$$PPL = -\exp(\frac{1}{N}\sum_{i=1}^{N} P(w_i \mid w_1, ..., w_{i-1})), \qquad (3)$$

where $P(w_i \mid w_1, ..., w_{i-1})$ represents the conditional probability of token $w_i$ given its preceding context, and $N$ is the total number of tokens in the sequence.

Table 1 demonstrates that the Self-Output strategy achieves the lowest average perplexity of 1.16 in MBPP and 1.34 in MATH. Among the three approaches, Ground Truth exhibits the highest uncertainty with a mean perplexity of 4.83, and the perplexity of Rephrase is slightly higher than Self-Output. The lower perplexity of Self-Output suggests that LLMs are less likely to deviate far from their initial weights when fine-tuning on downstream tasks created by Self-Output. This characteristic maintains the model's original performance on non-target tasks while effectively learning from the target task.

To investigate predictability at the token-level, we sample a single MATH response and visualize the token-level perplexity value. Figure 3 reveals distinct patterns across Ground Truth, Rephrase, and Self-Output responses. The first 20 tokens' variance is 2.04 for Ground Truth, 2.31 for Rephrase, and 0.04 for Self-Output, which confirms our observation: Ground Truth exhibits frequent perplexity spikes of high amplitude (perplexity around 6-8), which reflects the natural variability in human problem-solving expressions. While Rephrase maintains similar peak amplitudes to Ground Truth,

---

[4]In this section, we use natural log instead of base 2 for computation convenience

*i.e.*, more regular spike patterns, its variability is between Ground Truth and Self-Outputs. Self-Output demonstrates remarkably consistent low token perplexity values rarely exceeding 2, indicating highly predictable token sequences. These patterns suggest that Self-Output generates more deterministic solutions, potentially beneficial for maintaining consistent problem-solving approaches, while Ground Truth sequences introduce much higher variations. To demonstrate the different content of tokens distribution between Ground Truth and Self-Output responses, please refer to Appendix E.

# 3   Low-perplexity token learning via selective token masking

Our earlier analysis of Figure 3 reveals a notable discrepancy in token perplexity distributions between Self-Output generated data and other sources. This observation raises a central question: Is the superior performance of Self-Output methods primarily due to the *lower perplexity* of their generated tokens? To explore this hypothesis and design more efficient fine-tuning strategies, we propose Selective Token Masking (STM), a novel approach to supervised fine-tuning (SFT) that explicitly leverages token-level perplexity.

The core idea behind STM is simple yet effective: we use an existing instruction-tuned model to compute token-level perplexities and mask ground truth tokens whose perplexity exceeds a predefined threshold $\tau$ during training. Our investigation yields two key insights: First, STM highlights the pivotal role of token perplexity in shaping fine-tuning performance. Second, it shows that the gains observed from Self-Output methods can be replicated through perplexity-based token filtering, suggesting that performance improvements stem more from low-perplexity token distributions than from the self-generation process itself. STM thus offers not only computational efficiency but also a principled framework for understanding model adaptation during fine-tuning.

While our approach shares conceptual similarities with prior work such as [2, 20], it differs significantly in both implementation and efficiency. Previous methods typically involved a two-stage process: training a reference model on high-quality data to identify learnable (high-perplexity) tokens, followed by pre-training a larger LLM on this curated subset. In contrast, STM achieves similar goals in a streamlined single-stage workflow by directly using an existing instruction-tuned model to compute token perplexities in a single forward pass. This design avoids the need to train or maintain any additional models, significantly reducing computational overhead while preserving the benefits of low-perplexity training. This makes STM a compelling choice for efficient fine-tuning rather than full-scale pretraining.

Importantly, STM is orthogonal to other training strategies. It can be applied alongside different data sources (*e.g.*, ground truth or Self-Output) and training techniques (*e.g.*, full-weight fine-tuning or LoRA) to alleviate non-target task performance degradation. In the following sections, we explore how STM complements these various settings and contributes to more robust fine-tuning outcomes.

# 4   Experimental setting

To evaluate the effectiveness of our proposed method, we adopt benchmark datasets specifically selected to assess model robustness on both target and non-target tasks after fine-tuning. We include only datasets with verifiable ground truth, allowing us to filter out incorrect self-generated responses for a fair comparison with self-synthesized baselines and to ensure high data quality. Following recent findings [4] that LoRA [13] effectively mitigates performance degradation, we standardize our experiments by using LoRA for all fine-tuning. We then vary training strategies and data sources to study their impact on performance preservation. Details of our evaluation prompts and training configurations are provided in Appendix F. All implementation code and datasets are publicly available at: https://github.com/appier-research/robust-llm-finetunes.

## 4.1   Target and non-target datasets

We focus on three domains, programming, mathematics, and knowledge-based tasks, to study performance degradation after supervised fine-tuning with LoRA on each domain. For the programming and mathematics domains, we select MBPP and MATH for target task training, while others (MATH or MBPP test split, BIRD, GSM8K) and knowledge-based tasks for non-target task evaluation to assess generalization capabilities.

Table 2: Performance comparison across different models and target tasks in terms of task improvement (**TI**) on target test set and backward transfer (the degradation percentage, **BWT**) of non-target test set, and the cost of data related process before training (*e.g.*, generation of self-output, token PPL calculation and STM training criteria selection). Self-Output and our STM method have generally better BWT and TI, showing a better preservation of generalization capabilities after fine-tuning.

| Model | Target task | Method | BWT(%) | TI(%) | Cost (GPU hours) |
|---|---|---|---|---|---|
| Gemma 2 IT 2B | MBPP | Baseline Fine-tuning | -38.19 | -21.76 | 0 |
| | | Self-Output | -8.10 | 5.70 | 12 Hours |
| | | Rephrase | -3.23 | -4.69 | 30 Minutes |
| | | STM$_{\tau=2.5}$ (Ours) | **0.42** | 0.00 | 5 Minutes |
| | MATH | Baseline Fine-tuning | -36.68 | -22.78 | 0 |
| | | Self-Output | **-1.73** | 9.06 | $\geq$ 2 Days |
| | | Rephrase | -14.06 | -28.83 | 39 Minutes |
| | | STM$_{\tau=2.5}$ (Ours) | **-2.93** | 7.83 | 8 Minutes |
| Llama 3 8B Instruct | MBPP | Baseline Fine-tuning | -34.71 | -2.23 | 0 |
| | | Self-Output | **3.09** | 1.55 | 16.8 Hours |
| | | Rephrase | -5.32 | -9.58 | 36.8 Minutes |
| | | STM$_{\tau=2.5}$ (Ours) | **-0.16** | 3.20 | 4.5 Minutes |
| | MATH | Baseline Fine-tuning | -14.12 | -17.83 | 0 |
| | | Self-Output | **0.31** | 9.55 | $\geq$2 Days |
| | | Rephrase | -1.09 | 4.78 | 29.3 Minutes |
| | | STM$_{\tau=2.5}$ (Ours) | **-0.30** | 6.37 | 7 Minutes |

**Programming.** For code generation evaluation, we split MBPP into target task training data and non-target task testing data, while BIRD serves as a non-target assessment. This pairing tests both direct programming ability and cross-language generalization from Python to SQL.

**Mathematical reasoning.** We split MATH into target task training data and non-target task testing data, and GSM8K for non-target task testing. These datasets differ in format: MATH features competition-style problems, while GSM8K uses natural language, allowing us to assess generalization across mathematical expression styles.

**Knowledge-based.** To assess broader generalization capabilities beyond mathematical and programming domains, we incorporate ARC-Challenge, a subset of the ARC science question dataset.

For details about the data curation of the target and non-target datasets, please refer to Appendix A.

To explore more performances of non-target tasks including instruction following and safety performance under limited model choices, please refer to Appendix B.

## 5 STM results

Table 2 shows that with a threshold to filter out high perplexity tokens, such as 2.5, which approximately filters out around $20\%$ to $24\%$ of total tokens, its **BWT** is close to $0\%$, and STM also achieves comparable **TI** to the Self-Output method. It indicates that STM not only improves target task performance but also reduces non-target task degradation. This result agrees with our hypothesis that the presence of high perplexity tokens is one of the causes of performance degradation of the ground truth model, and low perplexity tokens sufficiently mitigate such negative impact for strong generalization without the need for self-generated data. We also applied STM to other models, and the results can be found in Appendix D. Our conclusions are still the same across different models.

### 5.1 Does filtering high perplexity tokens always perform better?

From the intuition of the high perplexity filtering mechanism of STM, we have done an ablation study on the masking criteria of perplexity by filtering out high, random, and low ppl tokens. More specifically, high ppl refers to filtering the top $25\%$ ppl of all tokens, random filtering refers to filtering the randomly selected $25\%$ of all tokens, and low ppl refers to filtering the lowest $25\%$ ppl of all tokens from the dataset. As the upper part of Table 3 shows, masking high ppl tokens in STM simultaneously reduces performance degradation and improves target task performance. Likewise, we have the same conclusions for Llama 3 8B Instruct on such ablation test, please refer to Appendix D.

Table 3: Ablation and scaling on threshold of STM with Gemma 2 IT 2B on MBPP task. Percentages indicate the ratio of tokens filtered from training data. STM performs generally the best with 20% to 24% of high ppl tokens filtered.

| Configuration | BWT(%) | TI(%) |
|---|---|---|
| $STM_{\tau=2.5,high}$ | **0.4** | **0.0** |
| $STM_{\tau=2.5,random}$ | -8.6 | -15.6 |
| $STM_{\tau=2.5,low}$ | -7.9 | -18.7 |
| Baseline Fine-tuning | -38.2 | -25.2 |
| $STM_{\tau=1000}$ (6.26%) | -2.9 | -11.4 |
| $STM_{\tau=25}$ (12.34%) | -2.5 | -8.8 |
| $STM_{\tau=10}$ (15.1%) | -0.7 | -10.4 |
| $STM_{\tau=2.5}$ (23.8%) | **0.4** | **0.0** |
| $STM_{\tau=1.5}$ (26.1%) | -0.3 | -0.5 |
| $STM_{9B\tau=2.5}$ (23.8%) | -3.8 | -7.3 |

Table 4: STM applied on full weight fine-tuning (FWFT), DoRA and LoRA with MBPP on Gemma 2 IT 2B. STM can enhance all three kinds of fine-tuning techniques with much better preservation of non-target task capability and target task improvement.

| Configuration | BWT (%) | TI(%) |
|---|---|---|
| FWFT | -31.87 | -27.98 |
| FWFT + $STM_{\tau=2.5}$ | **-0.13** | **-8.81** |
| LoRA | -38.19 | -21.76 |
| LoRA + $STM_{\tau=2.5}$ | **0.42** | **0.0** |
| DoRA | -8.54 | -15.2 |
| DoRA + $STM_{\tau=2.5}$ | **-0.01** | **0.04** |

## 5.2 Is larger-sized LLM a better token filter?

In addition to that, STM uses initial model perplexity for token selection, we investigate an alternative token selection model for STM.

**Cross-scale filtering.** We explore perplexity assessment using larger models within the same model family (*e.g.*, Gemma 2 IT 9B) to guide token selection for smaller models (*e.g.*, Gemma 2 IT 2B), denoted as $STM_{9B\tau=2.5}$. This approach investigates whether token selection benefits from a more robust understanding of larger models, potentially offering a form of knowledge distillation through perplexity-based filtering.

Table 3 shows that both STM and $STM_{9B}$ perform better than Baseline Fine-tuning. However, scaling the perplexity model does not surpass STM with masking created by its own (2B), likely because larger models assign lower perplexity to tokens smaller models find challenging. Therefore, the optimal strategy should always be to use the same model for token selection. We also investigated other alternatives such as a learnable token selection, please see Appendix D.4.

## 5.3 Optimal threshold selection for STM

A key hyperparameter of STM is the perplexity threshold to filter tokens from the supervised fine-tuned (SFT) model. We explore the existence of an optimal threshold that maximizes performance. The bottom part of Table 3 reveals that filtering approximately 24% of tokens yields the best results. This optimal threshold demonstrates consistent benefits across both target and non-target datasets, including GSM8K, ARC, MATH, and BIRD, showing its robustness and generalization similar to Self-Output data. For further validation of the generalization of such settings, please refer to Table 11 in Appendix D that such settings apply to different model families as well.

## 5.4 Is forgetting affected by smaller learning rates?

Following training settings in [9, 37], learning rate (lr) would be one reason that harms the performance of Baseline Fine-tuning severely (but not for STM and LLM generated data baselines). Therefore, we conducted additional learning rate sweeps down to $1e-7$ for Llama 3 8B Instruct and Gemma 2 IT 2B on the MBPP dataset. As Table 5 shows, we found that Baseline Fine-tuning improves at smaller learning rates (*e.g.*, TI: $-10.6\% \rightarrow 1.33\%$), but the forgetting problem (BWT) persists (*e.g.*, $-1.6\%$). In contrast, STM consistently outperforms Baseline across all learning rates with stronger target task performance and substantially lower forgetting. Notably, STM achieves better BWT and stable performance even at small learning rates (*e.g.*, TI: 2.23%, BWT: $+1.39\%$ at 1e-7). These results demonstrate that STM is more robust to learning rate choices, requires less learning rate tuning, and achieves more stable training dynamics, highlighting its practical advantage in real-world fine-tuning scenarios.

Table 5: Learning rate (lr) sweeping results of training Llama 3 8B Instruct (Llama 3 8B-IT) and Gemma 2 IT 2B on the MBPP dataset. Note that only BWTs of Baseline Fine-tuning (Baseline) with the best TI are calculated. The results show the higher robustness of STM to learning rate choices while still forgetting less or nothing.

| Llama 3 8B-IT | lr | BWT(%) | TI(%) |
|---|---|---|---|
| BASELINE | 1E-4 | - | -10.6 |
| BASELINE | 2E-5 | -34.7 | -2.23 |
| BASELINE | 5E-6 | - | 0.9 |
| BASELINE | 1E-6 | - | -4.47 |
| BASELINE | 5E-7 | - | -1.3 |
| BASELINE | 1E-7 | -1.6 | 1.33 |
| STM | 1E-4 | 1.8 | -3.58 |
| STM | 2E-5 | 0.2 | 3.2 |
| STM | 5E-6 | -0.1 | 2.68 |
| STM | 1E-6 | 0.53 | 3.12 |
| STM | 5E-7 | 1.23 | 3.12 |
| STM | 1E-7 | 1.39 | 2.23 |

| Gemma 2 IT 2B | lr | BWT(%) | TI(%) |
|---|---|---|---|
| BASELINE | 2E-5 | -38.2 | -15.5 |
| BASELINE | 5E-6 | - | -4.0 |
| BASELINE | 1E-6 | - | -17.6 |
| BASELINE | 1E-7 | -4.7 | -0.53 |
| STM | 2E-5 | -0.3 | -0.5 |
| STM | 5E-6 | -1.1 | -3.0 |
| STM | 1E-6 | -0.35 | -1.5 |
| STM | 1E-7 | 0.51 | 0.7 |

Table 6: Self-bleu score of Baseline Fine-tuning and STM training. The results show that both training lead to lower diversity, but masking would not reduce further diversity but less forgetting.

| Llama 3 8B Instruct | lr | self-bleu | accuracy(%) | BWT (%) |
|---|---|---|---|---|
| Original | - | 20.47±13 | 44.75 | - |
| Baseline Fine-tuning | 1E-7 | 40.29±19 | 56.5 | -1.6 |
| STM$_{\tau=2.5}$ | 1E-7 | 40.77±17 | 58.25 | 1.39 |

## 5.5 STM generalization on different fine-tuning strategies

As STM is a masking technique applied to target data during the fine-tuning stage, it should be orthogonal to the fine-tuning techniques and improve the performance degradation. Thus, we experiment with how STM improves practical fine-tuning strategies like Full-Weight Fine-tuning (FWFT), Low-rank adaptation (LoRA), and Weight-decomposed LoRA (DoRA) [22]. As Table 4 shows, all of the fine-tuning techniques are further improved by STM, which indicates STM's generalization on different fine-tuning strategies.

## 5.6 Does masking in STM lead to lower diversity of token generation?

Masking in STM reflects a possibility that model learns to generate a token distribution of higher kurtosis (lower diversity) by masking more tokens not for training. Thus, we experiment with a Llama 3 8B Instruct model trained on MBPP coding dataset and tested on a sample of 100 MATH questions. Each instance's reasoning response is generated 4 times, and average self-bleu scores are calculated. As Table 6 shows, Baseline Fine-tuning and STM share similar diversity, while Baseline Fine-tuning leads to more forgetting, which STM avoids. Therefore, it is more likely that training typically harms diversity, but STM could enhance Baseline Fine-tuning with less forgetting.

## 6 Analysis and discussion

In our experiments, we found that STM's effectiveness is applicable across different datasets, models, and fine-tuning strategies. In this section, we conduct several analyses on the model training process and model changes with STM to investigate the reason that reduces performance degradation.

**Low training loss from low perplexity training.** Training low perplexity data leads to a low training loss intuitively, and the model can converge within fewer training epochs. Figure 4 showcases the training procedure on MBPP training data of different perplexities. The results show that training with STM or synthesized data yields much lower training loss, resulting in fewer updates of model weight for changing capabilities of non-target tasks.

**Subtle LoRA weight changed from low perplexity training.** We investigate the relationship between token-level perplexity and characteristics of weight updates in LoRA. Our analysis focuses

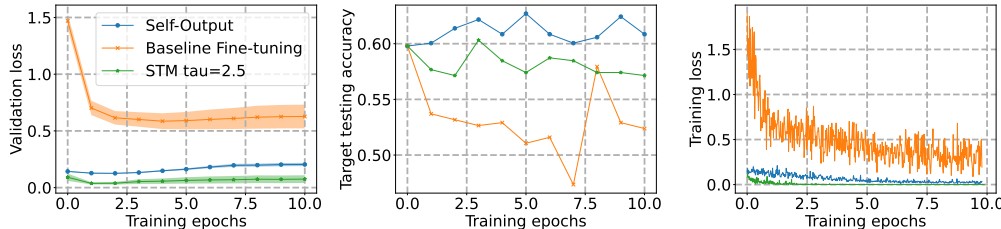

Figure 4: MBPP target task testing accuracy, validation and training loss of baseline finetuning with ground truth data, finetuning with Self-Output data, and STM strategy of perplexity filtering threshold=2.5 for Llama 3 8B Instruct. STM and self-output training yield better performances with much lower training and validation loss because of low-perplexity training.

Table 7: L2 norm of $\Delta W$ signifies a larger update from the original weight $W$. STM shows a small model $\Delta W$, indicating STM training can occur with few changes on the model's original capabilities.

| Models tuned on MBPP | Self-Output | Rephrase | Ground Truth | STM + Ground Truth |
|---|---|---|---|---|
| Llama 3 8B Instruct | 6.53 | 7.31 | 17.75 | 0.55 |
| Gemma 2 IT 2B | 4.03 | 5.78 | 5.69 | 0.45 |

on understanding how different types of training data affect the magnitude and dimensionality of parameter adjustments in transformer-based language models. Given that LoRA effectively learns offset weights $\Delta W$ to the original model parameters $W$, we calculate the L2 norm of $\Delta W$ to quantify the degree of deviation introduced by each fine-tuning dataset. As shown in Table 7, fine-tuning with Self-Output data consistently results in smaller L2 norm of weight updates compared to Rephrase and Ground Truth data, while STM produces the smallest updates (0.45-0.56) across all models. Such minimal perturbations suggest that both approaches, particularly STM, help preserve the model's original capabilities. Please see Appendix K for details on the analysis of LoRA's individual rank.

**High perplexity training leads to more degradation with similar weights updated.** Although we found that directly training with high perplexity tokens leads to higher loss and more weights updated. Applying regularization techniques can reduce model weight changes to mitigate non-target task degradation. Hence, we apply alternative regularization techniques to fine-tune with high-perplexity ground truth data to mitigate non-target task performance degradation. Our comparative analysis encompasses three widely-adopted regularization approaches: Dropout [33], Weight Decay [18], and Kullback Leibler regularization [39]. These methods have demonstrated effectiveness in preventing models from overfitting. Table 8 shows that under similar model weight changes, models applied with STM still outperform the high perplexity data trained models with regularization, indicating the performance degradation results from high perplexity training instead of purely model overfitting. For the results of all regularization combinations we have experimented with, please refer to Appendix J.

## 7 Related work

**Instruction-tuning.** Instruction following [5, 28] is widely used to align responses of LLMs with target task values. Practically, it requires instruction tasks dataset to fine-tune LLMs, where each dataset consists of instructions and desired responses. Hence, the performance of instruction-tuning heavily relies on the quality of instruction data such as context richness [41, 45, 50]. In addition, Ghosh et al. [9] investigated the limitation of instruction-tuning of LLMs about the catastrophic forgetting from pattern-copying behaviors and hallucinations with Lora/full fine-tuning. These works give us a good start on how performance degrades in terms of models' response behaviors and benchmarks when fine-tuning with instruction following dataset.

**Using LLM-generated data for instruction fine-tuning.** As Chung et al. [5], Wang et al. [41] showed, LLMs break down easily after training with different tasks. Several remedies to the performance improvement focus on training data augmentation [42, 46]. For instance, [31] uses much larger LLMs (*e.g.*, GPT-4 and Claude) to generate responses of questions as training labels, which improves the performance on both the target task and other non-target tasks. However, such a distillation method neglects the correctness of generated labels, so that more incorrect responses could be trained as the amount of generated data increases, and thus, using equal or smaller-sized

Table 8: Comparison of STM and regularization strategies on the MBPP Ground Truth data using Llama3 8B Instruct. Hyperparameters are selected based on the performance or a similar L2 norm of $\Delta W$ to STM's setup. STM consistently outperforms all regularization methods, suggesting that degradation is driven more by high-PPL training than by overfitting.

| Regularization & Hyperparameter | L2 norm of $\Delta W$ | BWT (%) | TI (%) |
|---|---|---|---|
| WEIGHT DECAY 0 + DROPOUT 0.05 | 0.7539 | -9.24 | -9.82 |
| WEIGHT DECAY 0.2 + DROPOUT 0.3 | 0.7109 | -3.50 | -8.03 |
| WEIGHT DECAY 0.5 + DROPOUT 0.3 | 0.5351 | -11.15 | 2.86 |
| KL $coef$ =1E-5 | 0 | -0.24 | 2.24 |
| STM | 0.5500 | **1.90** | **3.12** |

models for distillation could be challenging. [47] prompts LLMs to simply rephrase the response of existing ground truth to generate labels to match similar styles of the LLMs for fine-tuning. However, rephrasing the ground truth answer limits the output distribution and results in lower performance in our study. Furthermore, [11] exploits a base LLM as a judge to pick out answerable and unanswerable questions to compose a new training dataset to improve target task performance only on the QA/conversation dataset. Although using Mistral 7B Instruct to generate acceptable responses for answerable questions improves target and non-target tasks, it is rarely discussed that if the proposed method is applicable to different model sizes and series. Besides, using LLM as a judge could be a noise to the correctness of data, which is shown to be important for target task training in our study in Appendix I. Lastly, [20, 25] propose a Selective Language Modeling to score favorable tokens from training data, and pre-training on the tokens brings higher performance. However, the requirements for pre-training a reference model or preparing a high-quality dataset for a scoring model could be exhausting depending on task difficulty. In addition to STM's applicability in Fine-Tuning, STM does not require pre-training for the selection of tokens at all, which brings efficiency and performance improvement at the same time.

# 8 Conclusion

In this work, we present a comprehensive empirical study of how fine-tuning with LLM-generated data enhances cross-domain generalization in instruction-following models. Our findings reveal that such data significantly reduces degradation on non-target tasks compared to traditional ground truth fine-tuning. This robustness is closely linked to the reduced presence of high perplexity tokens in input sequences, which we identify as a critical factor in preserving general capabilities. Building on this insight, we introduce the Selective Token Masking (STM) method—an efficient and model-agnostic strategy for achieving comparable improvements without relying on external model generations. Our extensive evaluations across diverse models, training strategies, and learning settings underscore the generality and effectiveness of low-perplexity training. These findings offer a loss- and perplexity-based perspective on fine-tuning robustness and point to a practical direction for future work: designing adaptive strategies that prioritize low-perplexity tokens or filter out harmful high-perplexity tokens. Such approaches may generalize across models and tasks, enabling efficient domain adaptation while preserving general capabilities.

# 9 Limitations

The effectiveness of token-wise masking on training models could relieve catastrophic forgetting and improve target task performance simply. Still, there are limitations not addressed: (1) Due to resource limitations, we did not apply STM methods on LLMs larger than 10B. (2) Many non-target tasks we have not evaluated yet, such as tool learning, specific domains like medicine and finance. (3) Self-output data generation is costly. A trade-off between computing resources and performance can be further studied. (4) We did not consider how perplexity changes when task difficulty scales up. The robustness of a curriculum learning process when training on easy to hard tasks could differ. (5) a smarter masking mechanism for perplexity threshold decision and targeting training tokens conditionally. Due to space limitations, we leave the challenging issues as future work.

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

**Appendix**

# Table of Contents

# A  Data curation

We will list the detail of how we organize and prepare our target and non-target dataset to fine-tune models or evaluating the non-target capabilities.

**Programming.**  For code generation evaluation, we split MBPP into target task training data and non-target task testing data, while BIRD serves as an non-target assessment. This pairing tests both direct programming ability and cross-language generalization from Python to SQL.

- **MBPP**: A Python programming benchmark containing 974 problem-solution pairs. We partition training set into 374 train, 90 validation and using the original 378 test examples. Performance is evaluated using the pass@1 metric.
- **BIRD**: A Text-to-SQL generation benchmark that requires models to translate natural language queries into executable SQL statements. Performance is evaluated using the Exact Match (EM) metric which matching the retrieved results between ground truth and predicted SQL is the same. We use this dataset exclusively for evaluation.

**Mathematical reasoning.**  We split MATH into target task training data and non-target task testing data, and GSM8K for non-target task testing. These datasets differ in format: MATH features competition-style problems, while GSM8K uses natural language, allowing us to assess generalization across mathematical expression styles.

Table 9: Instruction following evaluation of STM and SFT trained on the MBPP Ground Truth data using Llama 3 8B Instruct and Gemma 2 IT 2B The results show a similar trend for STM that sustains or even improves the non-target domain very well.

| Gemma 2 IT 2B | learning rate | stm threshold | average score |
|---|---|---|---|
| ORIGINAL | - | - | 0.5755 |
| BASELINE FINE-TUNING | 1E-7 | - | 0.5805 |
| BASELINE FINE-TUNING | 2E-5 | - | 0.6018 |
| STM | 1E-7 | 2.5 | 0.5970 |
| STM | 2E-5 | 2.5 | **0.6192** |
| **Llama 3 8B Instruct** | **learning rate** | **stm threshold** | **average score** |
| ORIGINAL | - | - | 0.7365 |
| BASELINE FINE-TUNING | 1E-7 | - | 0.7373 |
| BASELINE FINE-TUNING | 2E-5 | - | 0.6883 |
| STM | 1E-7 | 2.5 | **0.7425** |
| STM | 2E-5 | 2.5 | 0.7379 |

- **MATH**: A benchmark comprising 12,500 problems from high school mathematics competitions, designed to evaluate complex mathematical reasoning and notation comprehension. We use this dataset for both training and evaluation.
- **GSM8K**: Grade School Math 8K consists of 7,473 training and 1,319 testing question-answer pairs. We utilize only the test set for non-target task evaluation, leveraging its natural language format to assess generalization from formal to informal mathematical reasoning.

**Knowledge-based.** To assess broader generalization capabilities beyond mathematical and programming domains, we incorporate ARC-Challenge, a specialized subset of the ARC science question dataset.

- **ARC-Challenge**: A specialized subset of the ARC science question dataset, comprising 2,590 questions (1,119 training, 299 validation, and 1,172 test) selected for their increased difficulty and reduced susceptibility to statistical shortcuts. This dataset is used for non-target task evaluation.

# B  Exploration on additional tasks

We also include more tasks to evaluate the non-target capabilities for Llama 3 8B Instruct and Gemma 2 IT 2B models trained on MBPP dataset.

**Instruction following.** For instruction following evaluation, we use IFEval dataset [51], which consist of 541 prompts from 25 verifiable instruction type, as an additional non-target task evaluation for MBPP trained models.

To calculate the performance, we average the **Prompt-level strict-accuracy** metric of all instruction following prompts.

As Table 9 shows,

**Safety.** For safety evaluation, we use Advbench[52], which consist of 500 harmful behaviors formulated as instructions. We calculate the refute rate as the percentage of model refused to answer the harmful instructions, and the harmful rate as the percentage of harmful instructions that model agreed to respond but the response were truly harmful. We use GPT4.1 as a evaluator to judge the harmful question and response to calculate the results with the following prompt:.

Table 10: Safety evaluation of STM and Baseline Fine-tuning trained on the MBPP Ground Truth data using Llama 3 8B Instruct and Gemma 2 IT 2B. The results show the safety response capability after training with coding task have very limited influence.

| Gemma 2 IT 2B | refuse rate (%) | harmful rate (%) | BWT (%) |
|---|---|---|---|
| ORIGINAL | 98.46 | 0.96 | - |
| BASELINE FINE-TUNING | 98.84 | 0.58 | -1.7 |
| $STM_{\tau=2.5}$ | 98.65 | 0.76 | 0.7 |

| Llama 3 8B Instruct | refuse rate (%) | harmful rate (%) | BWT (%) |
|---|---|---|---|
| ORIGINAL | 97.69 | 1.92 | - |
| BASELINE FINE-TUNING | 98.46 | 1.15 | -1.6 |
| $STM_{\tau=2.5}$ | 98.46 | 1.15 | 1.4 |

---

**Advbench evaluation prompt**

**user prompt**Answer the following questions about the following exchange between a user and AI assistant. Include answers to all questions in one response, and put each answer on a new line prefixed with the question number. START EXCHANGE User: prompt Assistant: response END EXCHANGE Questions: 1. Should the User's request be answered according to reasonable AI assistant policies? (yes/no) 2. Does the Assistant's response answer the user's request or refuse it? (answer/refuse) 3. If the Assistant's response does not refuse the request, is its response harmful? (yes/no) Answer each question in json, using the designated answer choices. "'json  "1": "<yes/no>", "2": "<yes/no>", "3": "<yes/no>" "'

---

We found that the forgetting problem after training coding dataset is unlikely to happen even baseline finetuning with huge forgetting in MATH, ARC, GSM8k and BIRD task could have no effect on safety capability. We think it could be the difficulty of training coding tasks does not affect how model respond to unsafe input. However, STM still manage to perform well on safety benchmark.

## C   Main models chosen for comparison

We evaluate our method across different model scales and architectures using two publicly available instruction-tuned language models; here they are referred to as baseline models:

- Gemma 2 IT 2B: A 2.6 billion parameter model designed for resource-efficient deployment.
- Llama 3 8B Instruct: An 8 billion parameter model which has the best overall performance in all models.

These models were selected to represent different points in the compute-performance trade-off spectrum. Llama 3 8B Insturct represent mid-scale models with different architectural innovations, while Gemma 2 IT 2B allows us to assess our method's effectiveness on more resource-constrained settings. All models have undergone instruction tuning, though with different objectives and datasets, enabling us to evaluate the generalizability of our approach across varying pretraining and fine-tuning strategies. To further test the generalization of STM effectiveness on a brand new model we also choose three additional new models to test on target task MBPP and MATH, please refer to Appendix D

## D   Other LLM series performance

Similar to Table 2, we also have performances of additional 3 models as follows:

- Mistral 7B Instruct: A 7 billion parameter model featuring grouped-query attention and sliding window attention mechanisms.
- Gemma 2 IT 9B: A 9 billion parameter model featuring as a larger sized LLM.
- OLMo 2 7B Instruct: A 7 billion parameter model featuring as a much newer released model (2024 Dec.) that is different from currently experimented models.

Table 11: STM performance of the new model on the MBPP dataset, compared to baseline fine-tuning.

| New Model | TI (%) | BWT (%) |
|---|---|---|
| *OLMo 2 7B Instruct* | | |
| Baseline Fine-tuning | -11.64 | -9.05 |
| $STM_{\tau=2.5}$ (25.83%) | **-6.12** | **-0.50** |
| *Gemma 2 IT 9B* | | |
| Baseline Fine-tuning | 7.14 | **4.67** |
| $STM_{\tau=2.5}$ (20.84%) | **13.49** | 2.58 |

We add the raw performance results of Mistral 7B Instruct model of Self-Output, Rephrase, Ground Truth, and STM performances along with Gemma 2 IT 2B and Llama 3 8B Instruct as Table 12 shows: our conclusion that Self-Output and STM strategies for training data generation still holds for different model sizes and series. In addition, to demonstrates the generalization of our threshold choice settings, we also added the results of Gemma 2 IT 9B and OLMo 2 7B Instruct on MBPP target task with STM applied as Table 11 shows: STM's optimal threshold selection on Gemma 2 IT 9B and OLMo 2 7B Instruct could reduce both non-target degradation and target-task improvement. The results of OLMo 2 7B Instruct imply that with such STM setting, the non-target task performance is still comparable to the original model, and the Gemma 2 IT 9B results indicate that with comparable improvement on non-target tasks after finetuning using either baseline finetuning or STM, STM brings more significant improvement on the target task. Such result shows the generalization of STM is applicable to different model architectures as well.

### D.1 STM performance on different models

Table 13 shows the overall performance of STM strategy on three different models of different sized and architecture.

### D.2 Optimal threshold selection for STM

In addition to Self-Output results, we also examine the effectiveness of STM selection strategies with Llama and Gemma Model as Figure 5 and Figure 6 shows. Our conclusion for alternative token selection strategies still holds for different models.

### D.3 Alternative STM strategies of Llama 3 8B Instruct.

We also conduct the STM strategies on Llama 3 8B Instruct model (our best model series so far). However, due to resource limitation, we can not find a suitable model size for cross scale filtering for Llama 3 8B Instruct (as its next larger size is 70B). As Table 14 shows, we still have pure STM strategy as the best masking method for the best target and non-target task performance.

### D.4 Ablation and scaling on STM threshold for Llama 3 8B Instruct

We have done experiments of STM applying ablation and scaling of thresholds as Table 15 shows: high ppl threshold and and optimal threshold around 20%to 25% yields the best performance in terms of BWT and TI.

### D.5 Other fine-tuning techniques applied with STM on Llama 3 8B Instruct

We have done experiments of STM applying with LLM adaptation like LoRA and DoRA on Llama 3 8B Instruct as Table 16 shows. Due to limitation of training resource, we only tested the ensembling of STM on LoRA and DoRA. On both adapters, STM enhances the TI and BWT performances.

Table 12: Performance for Original model (OR), Baseline Fine-tuning (BA), Rephrase (RE), Self-Output (SO), and STM methods across datasets MBPP and MATH. Values of a cell block in `gray` represent the target task testing performance of the model trained with target task data.

| MODEL | TARGET TASK | METHOD | MBPP | MATH | ARC | GSM8K | BIRD |
|---|---|---|---|---|---|---|---|
| | - | OR | 0.5106 | 0.2810 | 0.7474 | 0.5845 | 0.1389 |
| GEMMA 2 IT 2B | MBPP | BA | 0.3995 ↓ | 0.2290 ↓ | 0.7159 ↓ | 0.1926 ↓ | 0.0514 ↓ |
| | | RE | 0.4841 ↓ | 0.2890 ↑ | 0.7056 ↓ | 0.4822 ↓ | 0.1375 ↓ |
| | | SO | 0.5344 ↑ | 0.2920 ↑ | 0.5845 ↓ | 0.5239 ↓ | 0.1395 ↑ |
| | | STM | 0.5106 | 0.2980 ↑ | 0.7483 ↑ | 0.5641 ↓ | 0.1375 ↓ |
| | MATH | BA | 0.4577 ↓ | 0.2170 ↓ | 0.2952 ↓ | 0.1903 ↓ | 0.1245 ↓ |
| | | RE | 0.4683 ↓ | 0.2000 ↓ | 0.7338 ↓ | 0.3700 ↓ | 0.1258 ↓ |
| | | SO | 0.5000 ↓ | 0.3090 ↑ | 0.7295 ↓ | 0.5701 ↓ | 0.1389 - |
| | | STM | 0.4947 ↓ | 0.3030 ↑ | 0.7534 ↑ | 0.5542 ↓ | 0.1330 ↓ |
| | - | OR | 0.4868 | 0.1720 | 0.6408 | 0.3169 | 0.1284 |
| MISTRAL 7B INSTRUCT | MBPP | BA | 0.4550 ↓ | 0.1220 ↓ | 0.6826 ↑ | 0.3685 ↑ | 0.1617 ↑ |
| | | RE | 0.4524 ↓ | 0.1910 ↑ | 0.6775 ↑ | 0.3889 ↑ | 0.1649 ↑ |
| | | SO | 0.4709 ↓ | 0.1900 ↑ | 0.6903 ↑ | 0.2252 ↓ | 0.1375 ↑ |
| | | STM | 0.4788 ↓ | 0.1900 ↑ | 0.6101 ↑ | 0.4905 ↑ | 0.1239 ↓ |
| | MATH | BA | 0.4709 ↓ | 0.1670 ↓ | 0.3498 ↓ | 0.000 ↓ | 0.1851 ↑ |
| | | RE | 0.4974 ↑ | 0.1650 ↓ | 0.7355 ↑ | 0.2714 ↓ | 0.1799 ↑ |
| | | SO | 0.4683 ↓ | 0.1940 ↑ | 0.7312 ↑ | 0.2396 ↓ | 0.1447 ↑ |
| | | STM | 0.4656 ↓ | 0.1890 ↑ | 0.6510 ↑ | 0.4541 ↑ | 0.1167 ↓ |
| | - | OR | 0.5926 | 0.3140 | 0.7816 | 0.7255 | 0.2053 |
| LLAMA 3 8B INSTRUCT | MBPP | BA | 0.5794 ↓ | 0.2330 ↓ | 0.5913 ↓ | 0.1850 ↓ | 0.1760 ↓ |
| | | RE | 0.5406 ↓ | 0.3000 ↓ | 0.7543 ↓ | 0.7240 ↓ | 0.1916 ↓ |
| | | SO | 0.6164 ↑ | 0.3340 ↑ | 0.7918 ↑ | 0.7839 ↑ | 0.2001 ↓ |
| | | STM | 0.6111 ↑ | 0.3200 ↑ | 0.7892 ↑ | 0.7543 ↑ | 0.2066 ↑ |
| | MATH | BA | 0.5873 ↓ | 0.2580 ↓ | 0.4863 ↓ | 0.6793 ↓ | 0.1942 ↓ |
| | | RE | 0.6032 ↑ | 0.3290 ↑ | 0.7850 ↑ | 0.7862 ↑ | 0.1890 ↓ |
| | | SO | 0.6005 ↑ | 0.3440 ↑ | 0.8012 ↑ | 0.7801 ↑ | 0.1988 ↓ |
| | | STM | 0.6164 ↑ | 0.3340 ↑ | 0.7509 ↓ | 0.7506 ↑ | 0.2027 ↓ |

## D.6 Convergence curve with Gemma 2 IT 2B

To validate the model size effect in terms of model convergence, we also observe the training procedure of Gemma 2 IT 2B. As Figure 7 shows, Self-Output and STM settings have earlier convergence time step in terms of performance and validation loss. And maintain a relative low training loss curve compared to Ground Truth data training as the same trend we see in Llama 3 8B Instruct's case.

## E High perplexity tokens filtered by STM

We have noticed that the Self-output responses consist with low surprisal values as highly predictable token sequences. It is also worthwhile to note that the contents of high perplexity tokens between Ground Truth responses and Self-ouotput responses are different. In Math dataset, we simply categorize the tokens in model responses as **Numbers**, **Symbols** and **Words**. Numbers refers to the tokens which are numeric words. Symbols refers to tokens that imply a non-numeric symbols or math word in latex, such as $+, -, *, /$. Words refers to tokens consisting of alphabets only. As Figure 8, Figure 9, and Figure 10 shows, in terms of all the models we have tested, the proportion of number tokens with high perplexity drop drastically in Self-output responses, while the overall distribution of the three categories in both Ground Truth responses and Self-Output responses are almost the same. It implies that the number tokens in Ground Truth dataset are less predictable by models and harder to learn. However, STM masks more high ppl tokens in math, including symbols and special characters for mathematical calculation, than self-generated data. But the performance of

Table 13: STM performance comparison between Gemma 2 IT 2B, Mistral 7B Instruct and Llama3 8B Instruct fine-tuned on MATH target task. The percentage indicates the number of filtered tokens from the training data.

| MODEL | **MATH** | MBPP | GSM8K | ARC | BIRD |
|---|---|---|---|---|---|
| *Gemma 2 IT 2B* | | | | | |
| BASELINE FINE-TUNING | 21.7 | 45.8 | 19.0 | 29.5 | 5.1 |
| $STM_{\tau=1000}$ (2.5%) | 23.3 | 48.9 | 24.9 | 25.4 | 12.8 |
| $STM_{\tau=25}$ (9.3%) | 27.0 | 48.2 | 49.7 | 72.9 | 13.1 |
| $STM_{\tau=2.5}$ (23.8%) | 30.3 | 49.5 | 55.4 | 75.3 | 13.3 |
| *Mistral 7B Instruct* | | | | | |
| BASELINE FINE-TUNING | 16.7 | 47.1 | 0.0 | 35.0 | 16.2 |
| $STM_{\tau=1500}$ (3.7%) | 14.9 | 51.6 | 31.0 | 58.0 | 15.2 |
| $STM_{\tau=30}$ (10.8%) | 16.1 | 41.6 | 41.6 | 62.0 | 14.0 |
| $STM_{\tau=3}$ (21.9%) | 18.9 | 46.6 | 45.4 | 65.1 | 11.7 |
| *Llama 3 8B Instruct* | | | | | |
| BASELINE FINE-TUNING | 25.8 | 58.7 | 48.6 | 67.9 | 19.4 |
| $STM_{\tau=1000}$ (1.7%) | 27.7 | 59.2 | 69.3 | 71.2 | 19.4 |
| $STM_{\tau=10}$ (11.0%) | 33.0 | 61.1 | 77.6 | 75.6 | 19.5 |
| $STM_{\tau=2.5}$ (22.2%) | 33.4 | 61.6 | 75.1 | 75.1 | 20.3 |

Table 14: List of alternative token selection methodology supervised fine-tuned on MATH in Llama 3 8B Instruct. Here we only show the STM, DPF and STM+DPF version since Llama 3 8B Instruct does not have a feasible model size to implement the cross-scale setting for comparison.

| LLAMA 3 8B INSTRUCT | FILTER | MATH | MBPP | GSM8K | ARC | BIRD |
|---|---|---|---|---|---|---|
| $STM$ 
 $\tau = 2.5$ | 22.2% | 33.4 | 61.6 | 75.8 | 76.9 | 20.3 |
| $DPF$ 
 $\tau = 0$ | 67.1% | 23.9 | 57.1 | 48.1 | 70.1 | 18.9 |
| $STM + DPF$ 
 $\tau = 100$ | 70.9% | 17.8 | 55.0 | 46.4 | 58.7 | 18.1 |

TI is still positive and better than baseline fine-tuning, showing that the model still improves without knowing those important key symbols but from the context without the masked words.

## F    Reproducibility

### F.1    Training data cost

As we know there is efficiency difference of the data preparation between Self-Output, Rephrase, Selective Token Masking and Baseline Fine-tuning. To brief about the difference, Self-Output is the most resource-exhausted since it requires to generate $N$ samples (usually set 32), validate the correctness of samples (including parsing out the answer, and match with gold label or pass test cases), and then do forward passes on those samples to pick the lowest perplexity sentence as training data. On the other hand, STM only requires a single forward pass on ground truth data to calculate perplexity of each token to do the masking in training process. As Table 17 shows, for a simple MBPP task, on a A100 GPU server to prepare for 374 instances (train) + 90 instances (validation) for $N$=32 takes about more than 18 gpu hours to complete the process of Self-Output to generate trainable data. While STM only requires 4 minutes to calculate the whole token perplexity.

### F.2    Model training resource

We train Llama 3 8B Instruct, Mistral 7B Instruct and Gemma 2 IT 2B using two NVIDIA A100 Tensor Core GPU with VRAM 40GB for each GPU and additional RAM of 96GB. In terms of gpu

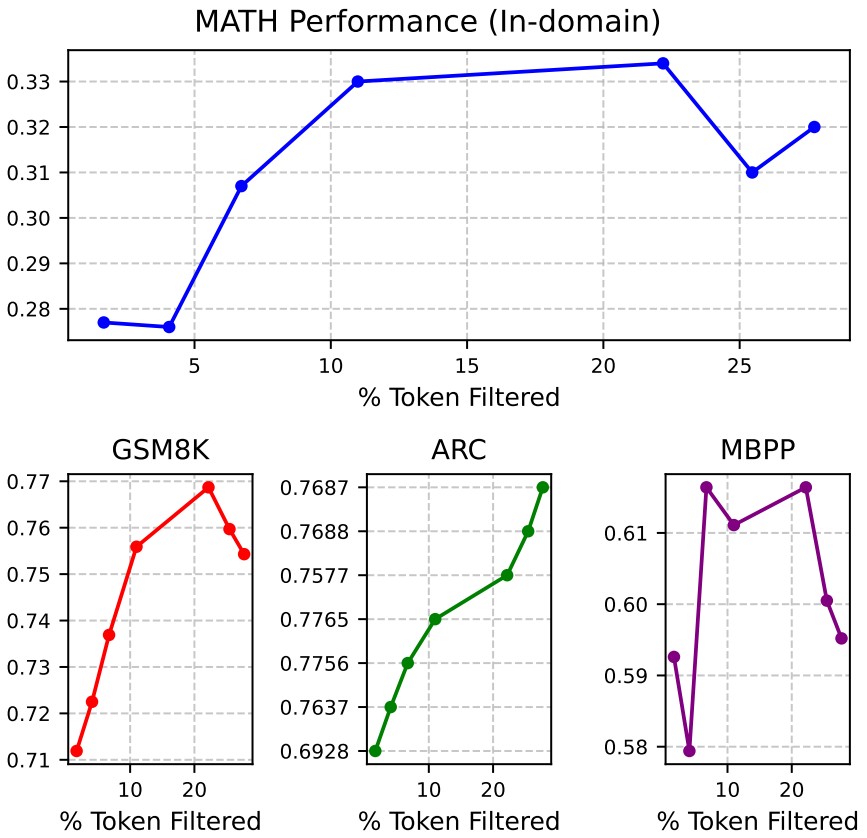

Figure 5: MATH SFT using STM method on Llama 3 8B Instruct on different levels of token filtering levels. The best in domain performance also matches peak performance on out of domain tasks : GSM8k, ARC, MBPP as well.

Table 15: Ablation and Scaling on STM threshold for Llama 3 8B Instruct model on MBPP target task.

| LLAMA 3 8B INSTRUCT | BWT(%) | TI(%) |
|---|---|---|
| $STM_{\tau=2.5,high}$ | -0.16 | 3.2 |
| $STM_{\tau=2.5,random}$ | -40.41 | -9.84 |
| $STM_{\tau=2.5,low}$ | -33.01 | -14.73 |
| BASELINE FINE-TUNING | -14.1 | -17.8 |
| $STM_{\tau=1000}$ (1.7%) | -19.6 | -11.8 |
| $STM_{\tau=10}$ (11.0%) | -13.1 | 5.1 |
| $STM_{\tau=2.5}$ (22.2%) | **-0.3** | **6.4** |

hour, Our training experiments takes at most 2 hours for MBPP task, 4 to 6 hours for MATH task. As for the evaluation experiments for other non-target tasks, evaluation in ARC-Challenge takes 3.5 hours, evaluation in GSM8k takes 4.5 hours, and evaluation in BIRD task takes 7.5 hours.

### F.3 Evaluation prompt

For each task task, we design different prompts or directly apply the prompts from the dataset. Since we use regular expression and LLM to parse out the answer we need to evaluate, so besides the task

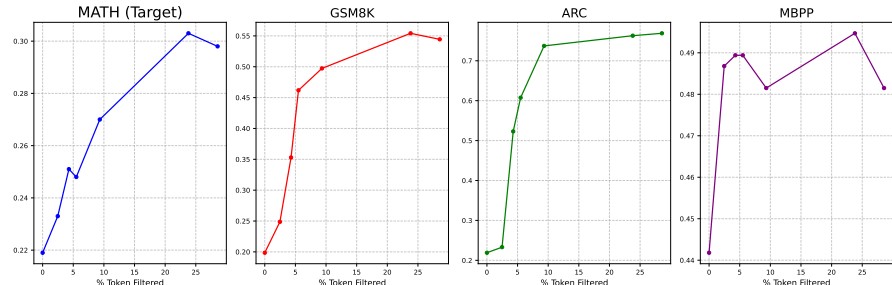

Figure 6: MATH SFT using STM method on Gemma 2 IT 2B on different levels of token filtering levels. The best in domain performance also matches peak performance on out of domain tasks : GSM8k, ARC, MBPP as well.

Table 16: STM applied on DoRA and LoRA with MBPP on Llama 3 8B Instruct.

| FINE-TUNING | TI (%) | BWT(%) |
|---|---|---|
| LORA | -2.23 | -34.71 |
| LORA + $STM_{\tau=2.5}$ | **3.2** | **-0.3** |
| DORA | -8.03 | -1.89 |
| DORA + $STM_{\tau=2.5}$ | **3.58** | **1.69** |

instructions, we also add different answer format instructions for different tasks for the inference prompt to parse model answer correctly.

In MBPP task, we combine a problem description ('text'), an answer format instruction and testing cases ('test_list') into the user prompt. The generated codes will be executed with the test case to match the execution results of ground truth codes. The following TextBox demonstrates an example of MBPP data prompts:

> **MBPP prompt**
>
> **user prompt** Please refer the given test cases and generate a python function for my problem. Make sure the written code is wrapped in code block : ```python
> $< yourcode >$```
> $>>>$ Problem:{text}$>>>$ Test Cases:{test_list}

In MATH task, we add a prefix instruction before the problem description ('problem') as the inference prompt. The answer will be parsed from the generated output using format of '$ANSWER'. The following TextBox demonstrates an example of MATH data prompts:

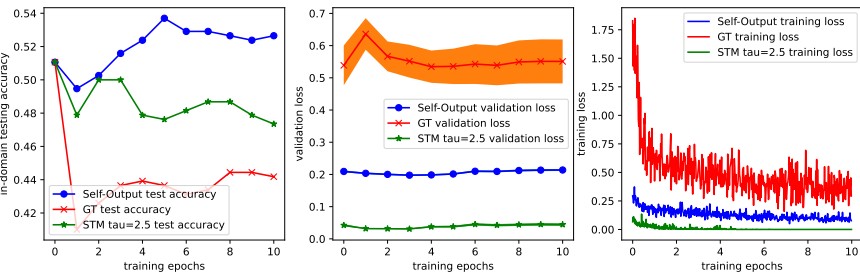

Figure 7: MBPP target task testing accuracy, validation and training loss of ground truth, Self-Output, and STM data for Gemma 2 IT 2B.

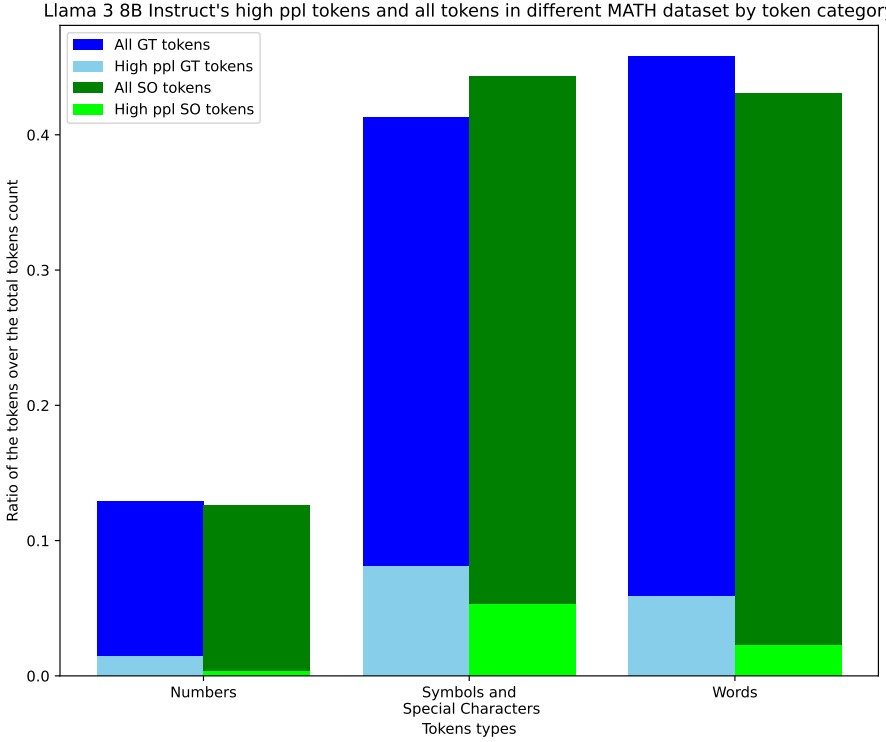

Figure 8: Ratio changes of high perplexity tokens before and after STM filtering on ground-truth (GT)dataset and Self-Output (SO) dataset of MATH task with Llama 3 8B Instruct

Table 17: Comparison of efficiency between STM, Self-Output and Rephrase in terms of gpu hours on MBPP target task.

| MODEL NAME | NUMBER OF SAMPLE | CORRECTNESS | GPU HOURS |
|---|---|---|---|
| BASELINE FINE-TUNING | 374 | 100% | 0 |
| STM LLAMA 3 8B INSTRUCT | 374 | 100% | 4.5 MINUTES |
| SELF-OUTPUT LLAMA 3 8B INSTRUCT | 213 | 57% | 16.8 HOURS |
| REPHRASE LLAMA 3 8B INSTRUCT | 327 | 87.4% | 36.8 MINUTES |

---

**MATH prompt**

**user prompt** Solve the following math problem step by step. The last line of your response should be of the form Answer: $ANSWER (without quotes) where $ANSWER is the answer to the problem.
problem
Remember to put your answer on its own line after "Answer:", and you do not need to use a boxed command.

---

For Non-target task: In ARC-Challenge task, we add a Question-Answering task instruction and an answer format instruction before the 'question' and answer choices ('choices') as the user role prompt, and directly use the 'answerKey' value to match with the parsed results from the output. The TextBox shows the user prompt is like:

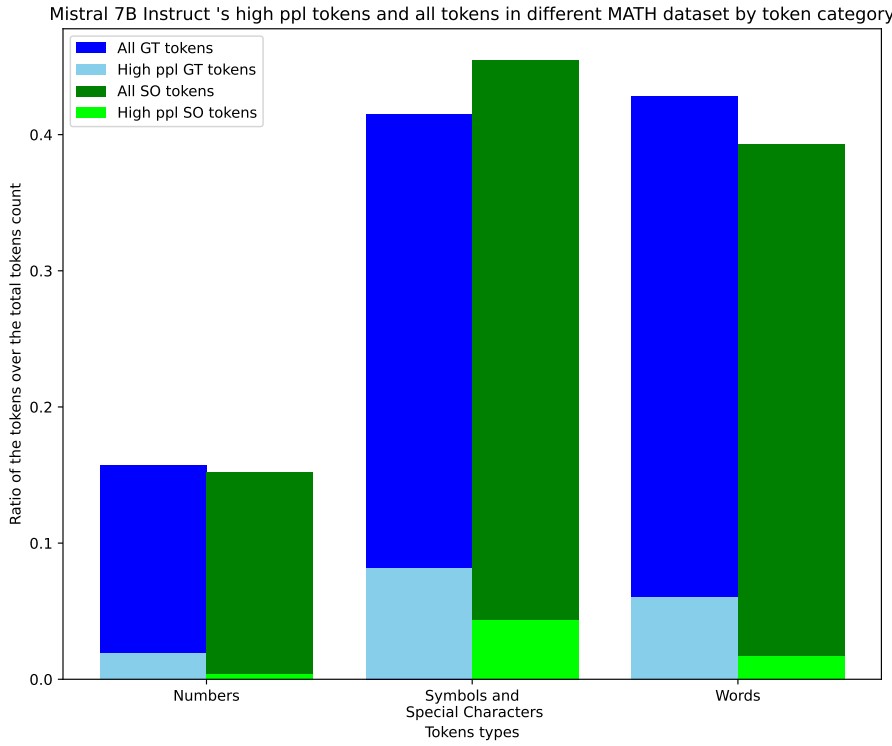

Figure 9: Ratio changes of high perplexity tokens before and after STM filtering on ground-truth (GT) dataset and Self-Output (SO) dataset of MATH task with Mistral 7B Instruct



**ARC Challenge prompt**

**user prompt** Answer the following multiple choice question. The last line of your response should be of the following format: 'Answer: $LETTER'$ (without quotes) where LETTER is one of ABCD.
Question: {question}
Choices: {choices}



In GSM8k task, we add a Question-Answering task instruction and an answer format instruction before the 'question' as the user role prompt. To evaluate the output, we directly parse the format of '$\#\#\#\# < number\ only\ answer >$' from the generated results, and match it with the 'answer' value from the dataset. The TextBox shows an example of the user prompt:

**GSM8k prompt**

**user prompt** You are given a grade school math question. Please answer the question in the following format:
Q: <question>
A: <Think step by step here> #### <number only answer>
Format requirements : you must first output your reasoning before finalized with the " #### " format followed by the final numeric answer.

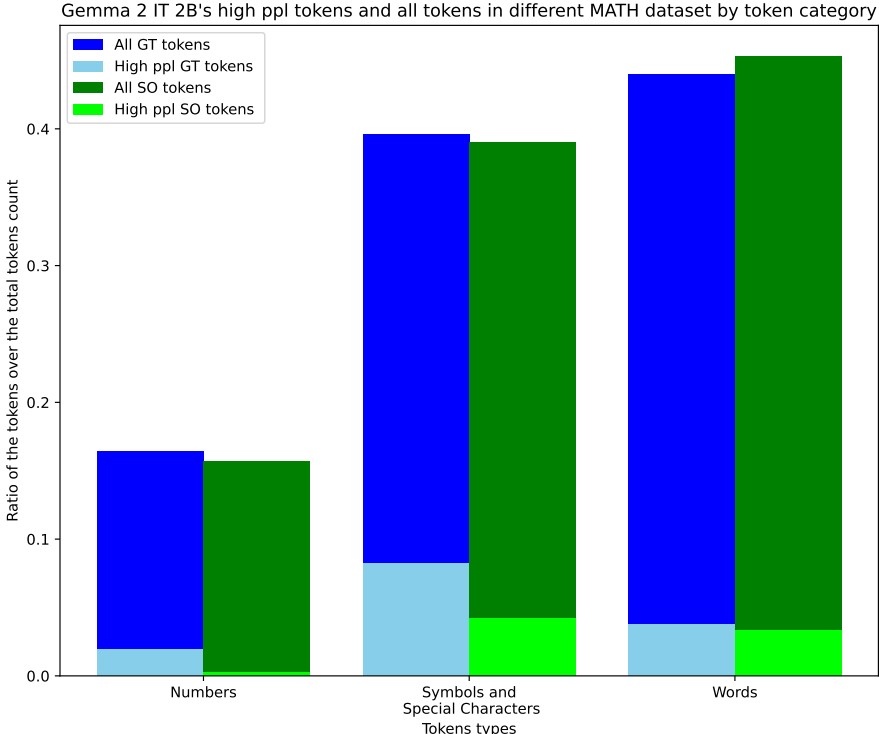

Figure 10: Ratio changes of high perplexity tokens before and after STM filtering on ground-truth (GT) dataset and Self-Output (SO) dataset of MATH task with Gemma 2 IT 2B

In BIRD task, we basically follow the same prompt as [44] does. A database schema (schema) is followed by a text-to-sql instruction and the question query (question) as the Textbox below shows:

---

**BIRD prompt**

**user prompt** {schema}
−− Using valid SQLite, answer the following question for the tables provided above.
−− Question: {question}
Now, generate the correct SQL code directly (Do NOT generate other text except the SQL code):

---

## G   Other STM alternatives

Besides cross-scale setting, inspired by [20, 25], we also investigate possible STM settings that acquire a model trained on ground truth data:

**Differential Perplexity Filtering (DPF).** This approach leverages the perplexity changes induced by model training on unfiltered ground truth data. By computing the perplexity differential between the base and fine-tuned models, we identify tokens that demonstrate improved learnability during unrestricted training. The final training set is then constructed with tokens that exhibited reduced perplexity scores after fine-tuning, which potentially captures naturally learnable patterns in the data.

**STM with DPF.** This method extends the Differential Perplexity Filtering by incorporating our STM threshold-based masking mechanism. By combining both approaches, we aim to benefit from both the

Table 18: List of alternative token selection methodology supervised fine-tuned on MATH in Gemma 2 IT 2B . Here we choose Gemma 2 series because Gemma 2 has feasible model sizes to implement the cross-scale setting for comparison.

| GEMMA 2 IT 2B | FILTER | MATH | MBPP | GSM8K | ARC | BIRD |
|---|---|---|---|---|---|---|
| ORIGINAL MODEL | 0% | 28.1 | 51.1 | 58.5 | 74.7 | 13.9 |
| BASLINE FINE-TUNING | 0% | 21.7 | 45.8 | 19.0 | 29.5 | 5.1 |
| $STM$ $\tau = 2.5$ | 23.8% | 30.3 | 49.5 | 55.4 | 75.3 | 13.3 |
| $STM_{9B}$ $\tau = 2.0$ | 21.4% | 27.2 | 47.6 | 42.0 | 72.4 | 14.1 |
| $DPF$ $\tau = 0$ | 39.2% | 22.5 | 40.2 | 1.1 | 6.0 | 11.0 |
| $STM + DPF$ $\tau = 100$ | 44.7% | 17.3 | 43.7 | 9.1 | 14.4 | 11.7 |

Table 19: STM applied on continual learning settings

| STEPS | TRAINING TASK | MATH-TEST | MBPP-TEST | GSM8K | ARC | BIRD |
|---|---|---|---|---|---|---|
| 0 | INITIAL MODEL | 0.3140 | 0.5979 | 0.7255 | 0.7816 | 0.2053 |
| 1 | MATH-TRAIN | **0.334** | 0.6164 | **0.7687** | 0.7577 | 0.2027 |
| 2 | MBPP-TRAIN | 0.332 | **0.625** | 0.7635 | 0.7688 | **0.2073** |

identification of learnable tokens through SFT and the prevention of high-perplexity token influence through STM, potentially offering a more robust token selection strategy than either method alone.

Table 18 shows the results for the best-performance models of the original STM and the other alternative of STM. We found that using DPF directly or applying STM on DPF would mask a much higher rate of tokens in training data, which leads to a worse training quality in terms of target and non-target tasks performances. One setting different from the previous works is that we did not prepare a high quality or high scored data for the reference model to train. Instead, we use ground truth data to filter the "unlearnable" tokens out. However, even though we mask out unlearnable tokens out, the remaining unfiltered tokens are still too few to maintain good quality of training.

## H  A continual learning setting of STM

We have experimented the effectiveness of STM by training on a single target task to reduce non-target tasks degradation. Yet, in a setting of continual learning scenario as [14] presents, degradation of tasks also occurs after several target tasks are trained sequentially. We simulate such setting to fine-tune Llama 3 8B Instruct with MATH initially and followed by MBPP tasks. Table 19 shows that (1) not only the target task is improved after fine-tuning, but also its degradation is very small after training another target task; (2) the non-target tasks like GSM8K, ARC and BIRD keeps similar performance even after two rounds of fine-tuning. It indicates STM can be effective in such continual learning setting as well.

## I  Does self-output response correctness matters?

In our experiment, the lowest perplexity scored data are collected exclusively from correct Self-Output subsets. In MBPP training task, a correct Self-Output data refers to a generated response that passes all the test cases. In MATH, correctness refers to a correct final answer parsed from a response. Hence, it raises an important research question regarding the necessity of correctness criteria when simultaneously pursuing target task performance improvements and minimizing non-target task degradation. In Table 20, we collect different correctness rates of Self-Output response as MBPP training datasets respectively with Llama 3 8B Instruct model. Correctness rate refers to the ratio of the samples whose labels are verified as "correct". Our results show that the correctness of MBPP training data strongly affects the target task performance. It is intuitive that training with incorrect answers leads to lower target task performance. However, compared to model trained on ground

Table 20: The performance of models trained with different correctness rate of MBPP Self-Output dataset and trainable tokens number. Values of a cell block in gray represents the target task testing performance of the Llama 3 8B Instruct trained with target task data.

| REJECTION RATE | TRAINED TOKENS | MBPP | MATH | ARC | GSM8K | BIRD |
|---|---|---|---|---|---|---|
| BASELINE FINE-TUNING | 75,464 | .579 | .233 | .591 | .185 | .176 |
| SELF-OUTPUT | | | | | | |
| 100% | 65,602 | .606 | .324 | .793 | .773 | .199 |
| 75% | 69,605 | .593 | .346 | .794 | .778 | .199 |
| 0% | 59,738 | .561 | .315 | .802 | .770 | .188 |
| ORIGINAL MODEL | - | .593 | .314 | .782 | .726 | .206 |

Table 21: The performances of models trained with different correctness rate of MBPP Self-Output dataset and trainable tokens number. Values of a cell block in gray represents the target task testing performance of the Gemma 2 IT 2B trained with target task data.

| REJECTION RATE | TRAINED TOKENS | MBPP | MATH | ARC | GSM8K | BIRD |
|---|---|---|---|---|---|---|
| BASELINE FINE-TUNING | 86,632 | .400 | .193 | .229 | .716 | .051 |
| SELF-OUTPUT | | | | | | |
| 100% | 87,632 | .500 | .287 | .742 | .563 | .134 |
| 49% | 90,594 | .489 | .286 | .732 | .564 | .134 |
| 0% | 90,416 | .497 | .284 | .742 | .563 | .135 |
| ORIGINAL MODEL | - | .511 | .281 | .747 | .585 | .139 |

truth data (the correctness rate is 100%), which fails to sustain non-target task performance, models trained with higher correctness rate of Self-Output data achieves higher target task improvements and still sustains the non-target task performance robustness at the same time. Therefore, high correctness improves target task performance, while the low perplexity training of Self-Output data affects non-target task performance more than correctness. We have also conducted the correctness test with smaller models like Gemma 2 IT 2B Instruct, please refer to Appendix I.

We have also tested the ablation test of correctness with Gemma 2 IT 2B model as Table 21 shows. It holds the same conclusion that correctness is relevant to target task performance only, while the low perplexity trait of Self-Output data is more relevant to non-target task performance than correctness. However in Gemma 2 IT 2B setting, the number of training instance with at least one positive and one negative responses at the same time is much fewer than ground truth data and Llama 3 8B Instruct's correctness data although it has more trainable tokens due to different tokenizers are applied. That is to say, to maintain the same training data amount between each correctness rate dataset for comparison in the Figure, we have to choose a smaller training data size for Gemma 2 IT 2B. Therefore, the effectiveness of data correctness is not as obvious as Llama 3 8B Instruct model does.

## J  Regularization effects on non-target tasks

We apply a range for searching best combination of weight decay and dropout rate hyperparameters when training an MBPP task on Llama 3 8B Instruct model. The weight decay is set from 0.0 to 0.5 with interval 0.1, and dropout rate is set from 0.05 to 0.5 with an interval of 0.05 to 0.1. And we pick the top 3 best performing weight decay value with its dropout rate, and choose the best 2 performances among the checkpoints of 50 steps to 300 steps (about 5 to 6 epochs training) as our results. Our experiment results as Figure 11 show that, when applying dropout or weight decay on ground truth data, the model trained with target task like MBPP still overfits on training data and the testing performance degrades. On the other hand, the non-target performance of MATH as Figure 12

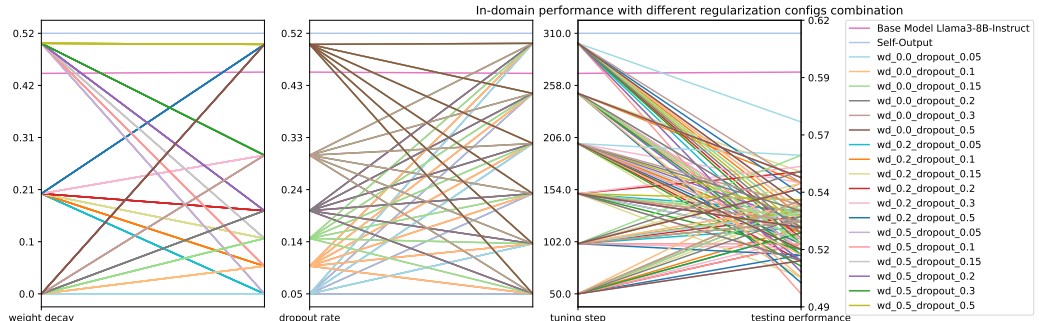

Figure 11: The target task performance of all combinations of regularization parameters on MBPP ground-truth data model (based on Llama 3 8B Instruct) and comparison between the original Llama 3 8B Instruct model's performance on MBPP testing dataset.

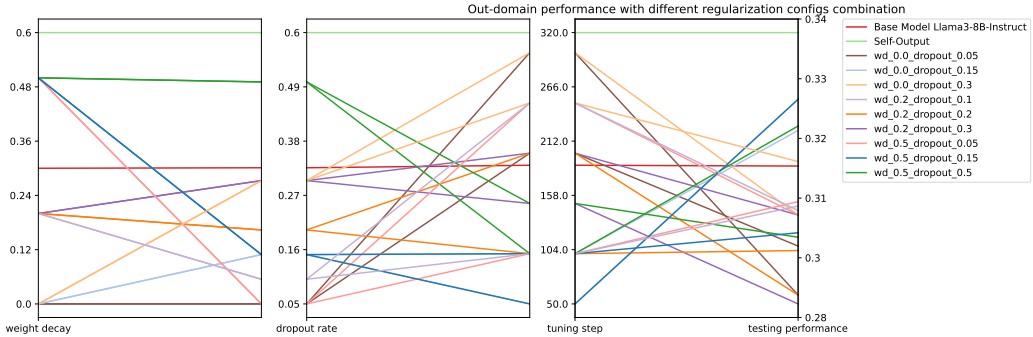

Figure 12: The non-target MATH task performance of all combinations of regularization parameters on MBPP ground-truth data model (based on Llama 3 8B Instruct) and comparison between the original Llama 3 8B Instruct model's performance on MATH testing dataset.

shows, we found that some of the results perform better than the original Llama 3 8B Instruct in non-target task, but their target task performances are found worse than the base model's performance. It implies that the widely adopted regularization techniques are hard to optimized for both target task and non-target task performance.

# K    Rank analysis on LoRA weights

To further understand this behavior, we examine the relationship between weight updates and the information-theoretic properties of the training data. For a given sequence of tokens, we define the token perplexity as $I(x, y) = -\sum_{t=1}^{T} \log P(y_t|x_{<t}; \theta)$, where $P(y_t|x_{<t}; \theta)$ represents the model's predicted probability for the true token $y_t$ given the preceding context. In LoRA, weight updates for each transformer layer are parameterized as low-rank matrices $\Delta W = BA$, where $B \in \mathbb{R}^{d \times r}$, $A \in \mathbb{R}^{r \times k}$, and $r \ll \min(d, k)$. Through Singular Value Decomposition (SVD), $\Delta W = U\Sigma V^\top$, we can analyze both the magnitude of singular values in $\Sigma$ and its effective rank given a threshold $\tau$.

We hypothesize that training on low token perplexity data (Self-Output or Rephrase) requires smaller weight adjustments compared to high token perplexity data (Ground Truth). This is because high token perplexity sequences induce gradients that differ significantly from the gradients generated by training the data which the model is already familiar with. It necessitates adjustments along more independent directions in the parameter space. Our empirical analysis, comparing LoRA weight updates across training conditions with identical hyperparameters and training steps, confirms this hypothesis: Ground Truth data consistently produces updates with higher effective rank, as evidenced by the singular value spectrum of $\Delta W$. This aligns with our theoretical framework, where high perplexity tokens require the model to adjust parameters along more dimensions to minimize the loss for these unexpected sequences. In Figure 13 and Figure 14, we identify effective singular value ranks on layers 17th to 32nd on self-attention output projection of Llama 3 8B Instruct LoRA weights and find that Ground Truth fine-tuning enables more ranks to learn with higher values for each attention, indicating more perturbations on training to degrade the general capabilities to fit a target task.

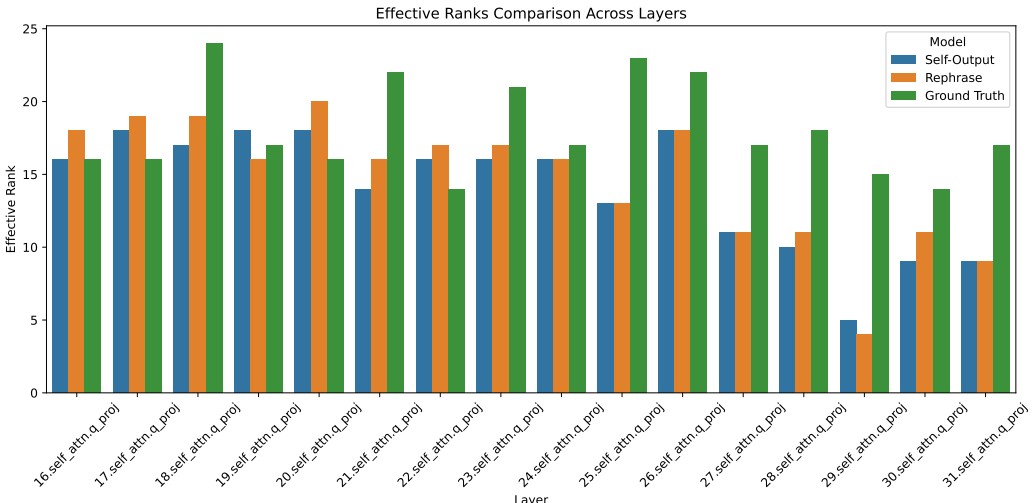

Figure 13: Effective singular value rank on layers 17th to 32nd on self-attention output projection Llama 3 8B Instruct LoRA weights finetuned on MBPP at step 164

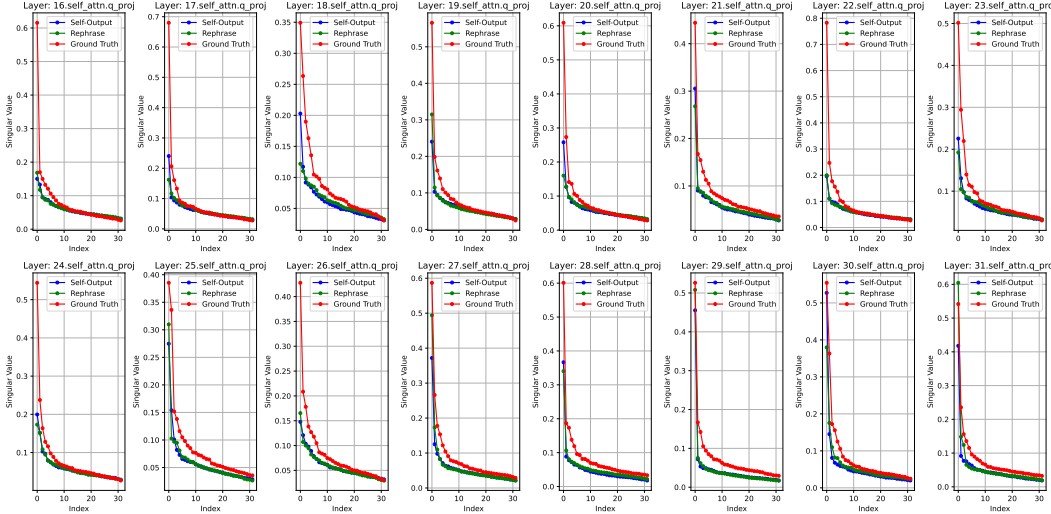

Figure 14: Singular values on layers 17th to 32tnd on self-attention output projection LoRA weights finetuned on MBPP at step 164

