# OpenReview forum: "Mitigating Forgetting in LLM Fine-Tuning via Low-Perplexity Token Learning"
_NeurIPS.cc/2025/Conference — NeurIPS 2025 poster_

### Official Review · Reviewer_C8Fr · 2025-06-24

**Clarity:** 4
**Significance:** 2
**Originality:** 2
**Rating:** 3
**Confidence:** 3

**Summary:**

This paper presents an empirical explanation for why generated data can help improve the down-stream performance of a fine-tuned LLM. In particular, the authors found that by systematically removing high-perplexity tokens from the fine-tuning dataset, the resulting down-stream model is able to achieve performance comparable with self-output based methods. Additionally, the authors provide STM, a simple method that filters out high-perplexity tokens from training data with a series of experimental analysis on differing methodologies.

**Questions:**

The reviewer notes the following questions for the authors:
- The reviewer feels there are risks with naively leveraging STM to remove high-perplexity tokens, is this not the case for specific applications of STM?
- Are there any challenges with selecting for STM threshold that would make the methodology more challenging in real-world applications?

**Ethical Concerns:**

["NO or VERY MINOR ethics concerns only"]

**Final Justification:**

Although the authors provided additional empirical evidence supporting STM, the limited theoretical justification behind an unintuitive methodology continues to ail the reviewer. Furthermore, comments raised by other reviewers regarding the experimental setup and the potential long-term consequences of STM during continuous training also raises valid concerns. As such, the reviewer will maintain their score for this paper.

**Limitations:**

Yes, the authors provided an summary of limitations and a short answer to if there are negative societal impact from this work.

**Paper Formatting Concerns:**

The reviewer has found no concern with paper formatting.

**Quality:**

2

**Strengths And Weaknesses:**

The reviewer notes the following strengths & weaknesses of the paper.

Strengths:
- This paper provides a compelling story with strong empirical results to justify the hypothesis of high-perplexity tokens in training data being a primary cause of performance degradation.
- The empirical evaluation spans multiple models and baselines used as points of comparison.
- The proposed Selective Token Masking (STM) is a simple and computational efficient method that filters out high-perplexity tokens from training data.

Weakness:
- The paper potentially provides a narrow and simplified view of high-perplexity tokens. For instance, some high-perplexity may be necessary during fine-tuning and STM may be truncating introducing instability in the fine-tuning due to unnatural gaps.
- There is limited theoretical justification for why low-perplexity tokens are more benign. A deeper theoretical analysis would help strengthen the argument provided in this work.
- Additional challenges with selecting for threshold may make STM less readily applicable in real-world settings.

---

> ### Author Rebuttal · Authors · 2025-07-30
>
> We sincerely thank the reviewer for the thorough and constructive comments. Please find the response to your questions below.
>
> > weakness 1. simplified view of STM cause training instable due to unnatural gap.
>
> Thank you for the insightful feedback. In our experiments with Self-Output and Rephrase, we observe that high-perplexity tokens are often rewritten into low-perplexity alternatives, leading to improved performance, lower training loss, and faster convergence, resulting in a smoother training curve compared to using ground truth data. Similarly, STM exhibits consistent trends in both training stability and performance.
> We believe that STM primarily filters out high-perplexity tokens that are unhelpful for training. Even when such tokens carry important information, the model is often able to recover it through contextual learning from the surrounding unmasked tokens. Our results on the MATH dataset support this: as shown in Appendix D, Figures 8, 9, and 10, STM removes a higher proportion of symbols and special characters compared to Self-Output, yet still improves both BWT and TI scores, demonstrating enhanced target-task performance and reduced forgetting.
> We agree that future extensions of STM could benefit from more nuanced selection mechanisms, e.g., conditioning on attention patterns or incorporating token roles (e.g., logical operators vs. distractors), to better distinguish between helpful and harmful high-perplexity tokens. We have added this direction to our discussion section as a promising avenue for future work.
>
> > Weakness 2. Limited theoretical justification
>
> Thank you for raising this important point. As noted in our limitations checklist, this work does not aim to provide a theoretical formulation and instead focuses on a comprehensive empirical investigation. To that end, we conducted extensive experiments across multiple model families, scales, datasets, and fine-tuning strategies (see Tables 2, 3, 4; Appendix C), all of which consistently demonstrate that low-perplexity token training, via STM or Self-Output, yields better preservation of general capabilities and improved target-task performance.
> Furthermore, in Appendix J, we analyze LoRA rank and weight updates and observe that STM and Self-Output generally result in lower-rank adaptations and smaller update magnitudes. This suggests that low-perplexity training tends to induce more localized updates, which may help retain pre-trained capabilities during task adaptation.
> While we do not provide a theoretical justification, we believe the consistent empirical trends observed across varied setups offer practical insights into the behavior of low-perplexity training. We agree that building a formal theoretical framework to explain these effects is an important direction for future work, and our findings may serve as a foundation for such efforts.
>
> >Weakness 3. Selection of STM make less steadily applicable in real-world application.
>
> We thank the reviewer for the insightful question. While selecting an appropriate threshold for STM can be challenging when deploying STM to entirely new domains, our empirical findings suggest that the current threshold setting (e.g., τ=2.5) is remarkably stable across a diverse range of real-world domains in our work: programming, mathematical reasoning, and factual knowledge retrieval. These domains are representative of practical applications where fine-tuning stability and generalization are critical.
> To further illustrate real-world applicability, we also apply STM to instruction-following tasks, IFEVAL. As shown below, STM consistently mitigates forgetting and improves performance relative to baseline fine-tuning across different learning rates and models:
>
> Instruction following (IFEVAL)
> |Gemma 2 2B-it | Learning rate | STM threshold |  Average score |
> | ---| --- | --- | --- |
> |    Original        |  -         |   -         |  0.5755            |
> |    basline ft      |  1e-7   |   -         |  0.5805            |
> |    basline ft      |  2e-5   |   -         |  0.6018            |
> |    STM            |  1e-7   | 2.5        |  0.5970            |
> |    STM            |  2e-5   |  2.5       |  0.6192            |
>
> | Llama 3 8B-it   | Learning rate | STM threshold |  Average score |
> | ---| ---| ---| ---|
> |    Original        |  -         |   -         |  0.7365
> |    basline ft      | 1e-7    |   -         |  0.7373
> |    basline ft      |  2e-5   |   -         |  0.6883
> |     STM            | 1e-7    |  2.5      |  0.7425
> |    STM             | 2e-5    |  2.5      |  0.7379
>
> While task-specific variability in perplexity distribution could affect optimal masking behavior in principle, our current result suggests that STM is already broadly applicable in practical settings. While we view more adaptive or data-aware thresholding mechanisms as a valuable future work, our results indicate that STM, as a simple and practical first step, already delivers robust performance across real-world fine-tuning tasks.
>
> > Q1. removal of high ppl token is a specific case for STM.
>
> To ensure STM is generalized enough, we take 5 different tasks, 5 different models ranging from 2B to 9B, originating from 4 different model families. And our conclusions still hold for such similar threshold selection based on the token filtering ratio. In addition to our scenarios, concurrent work of VLM (Liao et al, https://arxiv.org/abs/2504.15362) also has similar findings with generated data training in a multi-modal application, which shows STM’s possibility to apply in broader applications.
>
>
> > Q2. any challenge of threshold selection of STM.
>
> To address the challenge for generalization of threshold selection, we further apply our threshold selection on other new models we did not mainly discuss in this paper as well: OLmo 7B and Gemma 9B results in Appendix C Table 7, holding the same conclusion that STM benefits both target improvement and reduction in forgetting.
> It is still unknown how STM affects the learning when input contains a higher proportion of difficult, higher ppl context, e.g. specific domain data adaptation or hard-to-verify data, e.g. reasoning process, it makes us believe that considering not only tokenwise, but also attentionwise masking could benefit the model training as well, which is a potential future work to investigate.

---

> > ### Comment · Reviewer_C8Fr · 2025-08-04
> > **Response**
> >
> > I would like to thank the authors for their detailed response. Although I see the arguments made by the authors in regards to the empirical evidence supporting STM, I again hesitate to improve my score due to the limited theoretical justifications behind a methodology which both potentially weakens the down-stream model and is unintuitive.

---

> ### Author Response · Authors · 2025-08-04
> **Request for Clarification on Theoretical Justification and Intuition Concerns**
>
> Thank you for your thoughtful comment. We agree that theoretical justifications are important and appreciate your perspective. While we believe that our empirical findings offer meaningful contributions—particularly in the context of large language models—we would greatly appreciate it if you could clarify the type of theoretical justification you are referring to (e.g., convergence guarantees, optimality results, or another aspect).
>
> Additionally, we would be grateful if you could elaborate on what is meant by the method being “unintuitive.” A more detailed explanation would help us better understand and address your concern.

---

### Official Review · Reviewer_TbJZ · 2025-06-30

**Clarity:** 3
**Significance:** 2
**Originality:** 3
**Rating:** 2
**Confidence:** 4

**Summary:**

This paper addresses the problem of catastrophic forgetting in LLM fine-tuning, where specializing a model on a target task degrades its general capabilities. The authors empirically demonstrate that fine-tuning on LLM-generated data mitigates this forgetting compared to using ground-truth data, attributing this phenomenon to the significantly lower token perplexity of the generated sequences. Based on this key insight, they propose a simple yet effective method called Selective Token Masking (STM), which involves identifying and masking high-perplexity tokens within the original ground-truth training data. Through extensive experiments, they show that STM successfully preserves non-target task performance

**Questions:**

See weaknesses

**Ethical Concerns:**

["NO or VERY MINOR ethics concerns only"]

**Final Justification:**

I have revised my score from the original rating to 2. My main concern remains that the authors have not adequately explained why fine-tuning on the target task would lead to performance degradation, which raises questions about potential experimental flaws. In particular, the proposed method’s strategy of masking high-PPL tokens could prevent the model from learning sufficient knowledge for the downstream task. I would like to see a quantified analysis of how much this masking impacts performance, but the authors have not addressed this point convincingly in the rebuttal. If other reviewers could point out where I may have misunderstood, I would be happy to revise my score.

**Limitations:**

Yes

**Quality:**

3

**Strengths And Weaknesses:**

### Strengths

1. Writing is good. Readers can easily get what the authors are trying to convey.
2. The empirical finding on how high perplexity tokens affect LLM forgetting is intriguing and inspiring to the community.
3. Empirical analysis is thorough.

### Weaknesses

1. I am confused from the baseline fine-tuning results, where simple Baseline Fine-Tuning on the ground-truth data consistently yields a negative Task Improvement (Table 2). Logically, fine-tuning on the target task data should, at a minimum, improve performance on that same task, not degrade it relative to the base model. Am I missing something?
2. The paper's claim of mitigating cross-domain forgetting is not fully substantiated, as the "non-target" tasks remain within the same broad genre as the target tasks. For example, fine-tuning on MBPP (Python programming) is evaluated against BIRD (SQL generation), and fine-tuning on MATH is evaluated against GSM8K (math word problems). A more rigorous test of forgetting would involve measuring performance on truly disparate domains.
3. The core mechanism of STM introduces a critical and underexplored trade-off between preserving old knowledge and acquiring new, complex skills. High perplexity is a signal that the model finds a token surprising or uncertain, which is often the case for novel concepts, complex logic, or key steps in a difficult problem (e.g. a key piece of code for a hard coding problem). By explicitly masking these tokens, STM may prevent the model from learning the most challenging and valuable parts of the target task. This could place a ceiling on the achievable in-domain performance, which in many scenarios is a more detrimental outcome than the forgetting of non-target knowledge .
4. The paper's experiments are exclusively focused on reasoning domains (coding and math), which are characterized by logical structure and often have a single correct answer. The effectiveness of low-perplexity learning in these contexts may not generalize to more open-ended, subjective, or creative tasks. In domains like dialogue, summarization, or story generation, high-perplexity tokens can be crucial for stylistic variety, nuance, and creativity. Applying STM in such settings could inadvertently homogenize the model's outputs and hurt its generative capabilities, making the proposed method potentially unsuitable for a wide range of common LLM applications.

---

> ### Author Rebuttal · Authors · 2025-07-30
>
> We sincerely thank the reviewer for the thorough and constructive comments. Please find the response to your questions below.
>
> > Lower baseline finetuning results than original models.
>
> We appreciate the insightful concern. In related work [9], it discusses how instruction tuning’s copy pattern could possibly harm the performance, and could also affect the response quality of LLM. Such possibly “overfitting” problems could be solved either by carefully tuning the hyperparameters or other regularization methods. Here we provide carefully tuned baseline fine-tuning results with very small and not practically used learning rates in the previous rebuttal comment, which can improve the performance from the original model in terms of TI. While it still fails to surpass the performance of STM with a larger and commonly used lr. And STM sustains overall improvement of TI and BWT across learning rate, also showing a low sensitivity for learning rate with STM.
> In addition, in our work, we try to address the forgetting problem from fine-tuning, which is also mentioned by related work [5,35,36,40]. Thus, the positive TI metrics are not the main focus, but the BWT metrics. Yet we think it would be better to mention details about how fine-tuning results could suffer from easily overfitting/pattern-copy failure and catastrophic forgetting in our future version.
>
> > Only fintuning and evaluating on the similar domain at the same time
>
> Thank you for your concern. In section 4.1, we simply brief the target and non-target tasks and datasets used for experiments. We actually target to train on two datasets from two domains respectively, and test the rest of the four non-target datasets from three different domains altogether, as shown in Table 8. For example, we fine-tune on MBPP for the coding domain, but we evaluate not only against BIRD, but also the MATH-test, GSM8k, and ARC-challenge dataset together, and calculate the average BWT score for forgetting metrics calculation, and evaluate on MBPP-test for TI score for target task improvement calculation.
> We understand your concern about similar domains between training and evaluation, so we consider all three domains (programming, math, and knowledge-based QA) for evaluation at once under the suitable limitations ( which is the need for label-verifiable tasks for generated data baseline)  that fit our baselines.
> Furthermore, we believe that training on programming tasks with robust training like STM should ensure improvements on similar tasks as well. In many applications, this does not always hold for SFT; if in similar tasks SFT already forgets, then it will forget even more in truly disparate tasks than STM.
>
> > Masking prevent model learning from challenging and valuable parts.
>
> We understand the concern for STM to “underfit” some tasks with important key tokens being masked while training. Thus, we do have a simple trade-off study about the filtered ratios of STM threshold vs performance in Table 3 and the filtered token distribution study in Appendix D, Figures 8-10: STM masks more high ppl tokens in math, including symbols and special characters for mathematical calculation, than self-generated data. But the performance of TI is still positive and better than baseline fine-tuning, showing that the model still improves without knowing those important key symbols. We believe it is very likely that models can still learn novel concepts or logics based on the correct context we left in the training data, which, in our experiment, shows that STM has comparable performance with the self-output baseline. To address the difficulty of the problem, we could also explore curriculum learning (learn easy first, then hard later) to possibly avoid eliminating all higher perplexity tokens of challenging tasks. And we would like to emphasize that STM is built on top of SFT, and tasks that SFT could train on are also applicable for STM to address forgetting problems. Some open-ended tasks would not be suitable, such as RLHF or preference learning related tasks, or tasks with dynamic or unverifiable labels.
>
> > Unsuitable for the task of stylistic variety, nuance, and creativity:
>
> We agree there may be a possible limitation to consider forgetting training on only two domains. Thus, we also include a knowledge QA task like ARC-Challenge to evaluate the forgetting metric of models. Likewise, in the previous rebuttal, we believe most tasks SFT could train on should be covered by STM to address forgetting well. While in open-domain tasks including creative generation tasks, pure SFT may still underperform other methods due to there are few clear or gold standards of data that can be learned, but more preference learning strategies can work like RLHF, prompting, or LLM discussion (Lu et al)  solutions. We think it’s very important to consider how different traits/dimensions of high ppl training data differ when applying STM, and develop different masking strategies like conditional masking (a more finetuned masking way)  or curriculum learning (learning order of data by difficulty) as our future work to explore how to address learning with different traits of data. In this paper, we would like to start by figuring out that the root cause of forgetting is not purely because of generated data, but the high perplexity tokens in the ground truth training data.

---

> > ### Comment · Reviewer_TbJZ · 2025-08-04
> > **Response**
> >
> > I appreciate the authors’ detailed rebuttal. While it addresses some of my concerns, I still believe there are potential flaws in the experimental setup. In particular, I am still unconvinced by the explanation for the substantial performance drop on the in-domain dataset following baseline finetuning. I reviewed the related work, and my understanding is that it focuses on the general effects of IT, rather than its impact on the specific task for which it is conducted. The paper suggests that IT leads the model to adopt stylistic features from the training data. If that is the case, we would expect performance to improve on the in-domain dataset, while potentially decreasing on out-of-domain tasks. Moreover, this outcome appears to contradict standard methodologies for developing domain-specific LLMs. The conventional approaches would use continual pretraining combined with IT, or direct IT on its own, depending on the amount and type of data available for downstream task. A degradation of performance on the very dataset used for tuning is therefore a significantly counterintuitive result.

---

> ### Author Response · Authors · 2025-08-04
>
> We acknowledge your concern about the performance drop on in-domain datasets, in our other rebuttal `poeA`, we have conducted additional sweep of learning rate, which reveals that the initial performance drop was indeed partially due to suboptimal hyperparameter selection. With appropriate learning rate baseline finetuning can achieve positive gains on in-domain tasks (+1.33% on MBPP).
>
> However **when compared with STM methods, STM consistently outperforms baseline fine tuning on in-domain tasks**. This demonstrates that STM is not only more effective but also more robust to hyper-parameter choices.
>
> MBPP results :
>
> | Llama 3 8B | lr      | TI | BWT |
> | -- | --- | --- | --- |
> | baseline ft  | 1e-4 | -10.6%|   - |
> | baseline ft  |2e-5 (reported in paper)  | -2.23%|-34.7%|
> | **baseline ft**  | **5e-6**  | **0.9%**   |   - |
> | baseline ft  |1e-6  | -4.47%|   - |
> | baseline ft  |5e-7  | -1.3%  |   - |
> | **baseline ft**  |**1e-7**  | **1.33%** | **-1.6%** |
> |     **STM**       | **2e-5**   | **3.2%**  |  **0.2%** |
> |    **STM**        | **5e-6**  |  **2.68%**|  **-0.1%** |
> |    **STM**        |**1e-6**  |  **3.12%**|   **0.53%**|
> |    **STM**        |**5e-7**  |  **3.12%**|  **1.23%**|
> |    **STM**        |**1e-7**  |  **2.23%**|  **1.39%**|

---

> > ### Author Response · Authors · 2025-08-08
> >
> > Thank you for your thoughtful review and the time you have dedicated to evaluating our paper. We have added a detailed comment (results from the previous rebuttal session to another reviewer) addressing your recent concern and would be grateful for the opportunity to clarify any remaining questions.
> >
> > We warmly invite you to revisit our response, as your feedback is invaluable for further improving the clarity and quality of our work.

---

### Official Review · Reviewer_PAG5 · 2025-07-01

**Clarity:** 3
**Significance:** 2
**Originality:** 2
**Rating:** 4
**Confidence:** 4

**Summary:**

This paper investigates using LLM-generated data for finetuning when performing cross-domain generalization. The paper systematically investigates the augmentation of a fine-tuned language models onto a new task or domain by finetuning on human-written data, LLM-rephrased outputs of human written data, and the self-output of an LLM that is semantically equivalent to the human written responses. In their analyses, they find that model degradation on existing tasks is correlated with the average sentence perplexity of data points in all three methods. Thus, the authors utilize a new method that selectively masks tokens with high perplexity in the training data. In their experiments, they find that their proposed method is comparable to the best performing baseline of "self-output", while being much more computationally efficient, yielding a cheaper and effective alternative to domain expansion.

**Questions:**

Compared to the programming dataset, STM in the MATH dataset consistently (across models) yields lower performance compared to Self-Output. Are there specific reasons that this dataset or type of reasoning is not as effective when using lower perplexity?

How does the model performance, after training with STM, fare on more general tasks, like MMLU, Safety, and Instruction following? These results would further give insight on the proposed methodology.

Further investigations on the threshold of STM for general diversity, brevity, and utility of LLM generated text would also be insightful experiments to evaluate, especially for proving the safety and effectiveness of this low-perplexity training method.

The organization of the paper needs some improvement. Some references to figures across the paper are jarring and seem out of place. The general organization also resembles the discovery process in the paper (i.e. running the baseline, finding a trend, applying a new method to address the trend), which could be improved significantly for better clarity.

**Ethical Concerns:**

["NO or VERY MINOR ethics concerns only"]

**Final Justification:**

The paper is explores the idea of removing high perplexity tokens for better domain adaptation. The rebuttal by the authors includes additional ablation studies and suggestions of future domains or tasks to explore their STM method. I will be maintaining my score as these proposed changes do not strengthen the paper enough to be an "accept".

**Limitations:**

No, there should be further exploration on the safety and helpfulness of LLMs after cross-domain generalization. Only a few types of domains and tasks are explored (i.e. programming and mathematical reasoning).

**Quality:**

3

**Strengths And Weaknesses:**

Strengths:
Detailed experiments for cross-domain generalization. This paper does a good job at investigating existing baselines for using LLM-generated data for finetuning, analyzing results and finding a correlation between high perplexity tokens and model degradation, and developing a new method for selectively masking out high perplexity tokens on human written data to achieve better performance on mathematical and programming tasks.

The proposed method poses a very low computational constraint while being competitive with existing baselines. While their method does not show a clear improvement over existing baselines, their method is orders of magnitude less computationally expensive to run than the best baseline model of "self-output". This is especially important as the number of tasks or domains increases, enabling faster and cheaper cross domain generalization.

Weaknesses:
There are no analyses on the potential degradation of models on training on low perplexity data, especially over a long period of time. Intuitively, by removing all high-perplexity inputs when training, the model's diversity in responses will degrade over time. This investigation of model diversity remains unexplored, as the authors only explore task correctness. However, key papers like (Wang et al., 2023) hypothesize that the diversity of model outputs can correspond to different chains of thought that result in the correct answer, indicating that the diversity can result in improved reasoning capabilities. This is also more likely to be an issue for harder questions and evaluation datasets, as open-ended or difficult questions may require different solutions. As such, this paper does not explore the impact of decreased diversity of LLMs, especially over multiple iterations of finetuning, and the impact of high perplexity on difficult or rare reasoning questions is also not explored.

They experiment only on programming and mathematical datasets. While this is mentioned in the limitations, an exploration of finetuning on reasoning datasets by removing perplexity tokens and evaluating on general safety and language tasks like MMLU is crucial. Overall, the investigation of the full impacts of training on only low perplexity tokens remains unexplored in this paper.

---

> ### Author Rebuttal · Authors · 2025-07-30
>
> We deeply appreciate the reviewer’s insightful and detailed feedback. Below, we address each of your questions in turn.
>
> > Weakness 1. training over time and lower diversity with STM
>
> Thank you for the insightful concern. Although our method manages to address forgetting after supervised fine-tuning instead of diversity, we still understand the reviewer’s concerns about the difficulty of problems and diversity over multiple training. We think the difficulty distribution of problems is an unexplored topic of STM training and a great future work to do.
> For multiple times of training sessions, in Appendix G, Table 15, we have investigated a simple scenario of STM applied to continual training, which is training on a model with different datasets sequentially. The results showed that forgetting is also mitigated during each turn of training and enhances the performance of related domains as well.
> Based on the reviewer's suggestion, we also observe how diversity trades off with forgetting after training the model on the MBPP coding dataset. The experiment is tested on 100 MATH questions. Each instance’s reasoning response is generated 4 times, and average self-bleu scores are calculated. The results showed that even with a similar diversity (self-bleu), SFT results in more forgetting,  which also emphasizes STM’s capability to mitigate forgetting. We understand the reviewer’s concern that training could reduce diversity, but we think it is not caused by STM, as the diversity is almost the same with pure SFT. However, we believe that STM should focus on non-open domains that SFT could apply to, which have verifiable answers for models to train on, so that STM could enhance SFT with less forgetting.
>
> Proposed experiment on math reasoning x100 with MBPP trained Llama3 8B-it models
> | model + lr     | self-bleu (%)|accuracy(%) | BWT |
> | --- | --- | --- | ---|
> | original         |20.47±13|44.75  |    -    |
> | baseline1e-7|40.29±19| 56.5  | -1.6%|
> | STM lr=1e-7 |40.77±17|58.25 |1.39%|
>
> > Limiation: exploration on reasoning dataset, including instruction following, safety and MMLU.
>
> We thank the reviewer for the critical insight.  We think it’s very crucial to explore more tasks that are not covered in our work, as our limitation section. We also experimented with a non-target knowledge task (arc) other than just programming and mathematical datasets for forgetting dataset metric comparison (BWT). And reasoning dataset is also a great direction to investigate, while in this paper, to compare with baselines like self-output and rephrase, reasoning dataset is not easy to construct because we do not have a correct evaluation for the generated reasoning data, which could bring noise to the training as well. Because our core concept is to figure out that forgetting stems from high perplexity tokens instead of purely model-generated data, we need to compare generated baselines with applicable datasets. We believe tasks like knowledge QA, reasoning, safety, or instruction following tasks could be great future works to explore STM’s training effectiveness as well.
> Here we prepare a simple test for Instruction following on Gemma2-2B-it and Llama3-8B-it. We can still see that even though small lr prevents instruction following capability from decreasing, STM manages to maintain and even improve such capability better.
> Instruction following (IFEVAL)
> |Gemma 2 2B-it | Learning rate | STM threshold |  Average score |
> | --- | --- | --- | --- |
> |    Original        |  -         |   -         |  0.5755
> |    basline ft      |  1e-7   |   -         |  0.5805
> |    basline ft      |  2e-5   |   -         |  0.6018
> |    STM            |  1e-7   | 2.5        |  0.5970
> |    STM            |  2e-5   |  2.5       |  0.6192
>
> | Llama 3 8B-it   | Learning rate | STM threshold |  Average score |
> |--- | --- | --- | --- |
> |    Original        |  -         |   -         |  0.7365
> |    basline ft      | 1e-7    |   -         |  0.7373
> |    basline ft      |  2e-5   |   -         |  0.6883
> |     STM            | 1e-7    |  2.5      |  0.7425
> |    STM             | 2e-5    |  2.5      |  0.7379
>
> As for exploration in safety,  we have tested the results on the safety benchmark (advbench). Due to the limited training domains of programming and coding only, we found that neither baseline fine-tuning nor STM would cause degradation in the safety benchmark (advbench), which likely means that the difficulty of training tasks or related capability after training does not affect the performance on safety response.
>
> > Q1. lower improvement of STM than self-output method in math (reasoning)
>
> We thank the reviewer for raising the insightful concern. First of all, we believe the lower perplexity token trait of self-output data benefits the training to reduce or even improve forgetting. Second, we have initially observed some filtered tokens case study for the MATH task in Appendix D, figures 8,9, and 10. It does show that compared to self-output data, STM’s MATH ground truth data intrinsically filters out higher proportions of high perplexity symbolic and special character tokens, which could be important logics or reasoning processes, resulting in lower TI. While in BWT the averaged scores differ in less than 1% of the original scores, which could result from randomness of inference results of LLM and indicate a good preservation of generalization for both STM and Self-output.. To address the lower TI issue, we believe STM with conditional tokens masking could be a potential direction to explore when models could learn which high perplexity and “non-functional” tokens to filter.
>
> > Q2. STM effectiveness on more general tasks like instruction following, safety or MMLU
>
> We appreciate the concern raised by the reviewer. As we mentioned in the Limitation section, to indicate the forgetting stemming from low perplexity, we managed to compare the approaches that also bring low perplexity training data: Self-output and rephrase. However, these approaches are limited when labels of generated results are hard to evaluate correctly. Besides, we tested ARC for a knowledge QA based dataset; we do believe STM-trained models also mitigate forgetting in general tasks like MMLU, Safety, and Instruction following, since these tasks intrinsically contain high perplexity tokens to filter. Therefore, there is also a great potential to investigate STM effectiveness in those datasets.
> For further study, please see the previous comment about the results in the instruction following task. The trend of instruction following meets our expectations. And the comments we left about the safety benchmark in the previous comment as well.
>
> > Q3. further investiagtion on the trheshold selection of STM cause on diversity, safety
>
> Thank you for the insightful feedback. We have an initial study about high ppl tokens distribution analysis in the MATH dataset and the generated dataset in Appendix D, Figures 8~10. In the analysis, we find a different proportion of high ppl in the generated dataset and ground truth data. But removing the high ppl tokens in ground truth data for STM training does not harm the target task and non-target task performance. To address more aspects like diversity, please see the previous initial study on self-bleu diversity on math reasoning. And for the safety issue, as we mentioned in weakness 2. Safety may not be a forgetting issue for model training with coding and math domains. STM and baseline fine-tuning rarely forget safety in advbench in our case.

---

> ### Comment · Reviewer_PAG5 · 2025-08-07
>
> I would like to thank the authors for their detailed response. These additional experiments and proposed methods would strengthen the paper further. I will be maintaining my score.

---

> > ### Author Response · Authors · 2025-08-07
> >
> > Thank you very much for your evaluation and encouraging feedback. We sincerely appreciate your thoughtful engagement throughout the review process. We're glad that the clarifications in our rebuttal helped address your concerns.
> >
> > We also appreciate your suggestions regarding the extra results on diversity and Instruction following dataset, and we will incorporate these improvements into the revised paper.

---

### Official Review · Reviewer_poeA · 2025-07-07

**Clarity:** 4
**Significance:** 3
**Originality:** 3
**Rating:** 5
**Confidence:** 4

**Summary:**

This paper focuses on the challenge of preventing performance decrease on other tasks when performing domain-specific supervised finetuning for large language models. In particular, it explores the central idea that finetuning with high perplexity tokens (as measured by the model being finetuned) leads to more forgetting. The authors begin by demonstrating that this occurs empirically and then propose several methods for avoiding such high-perplexity tokens. The first is to replace the ground truth responses in the SFT dataset with either the model's own response to the prompt (Self-Output) or its rephrasing of the output (Rephrase). While self-output is shown to lead to lower average perplexity, it is computationally more expensive and requires reliably verifiable outputs since multiple generations must be performed to obtain a correct response. The second is selective token masking (STM) in which tokens above a given perplexity threshold are masked out during training. This is done both with the model's own perplexity and with the perplexity given by a larger model in the same family. The efficacy of all these methods is empirically explored in code and math SFT with the target tasks MBPP and MATH respectively.

**Questions:**

1. The paper describes the Rephrase method as advantageous compared to Self-Output given it does not require reliably verifiable outputs. Is any checking done with the Rephrase data to ensure it matches the solution of the ground truth?

2. You explore finetuning 2B and 8B models in this work. Have you looked at the perplexity of the ground truth response for larger models (say a 70B model)? In considering how these methods scale, it would be useful to know whether you start with the same underlying scenario where the ground truth SFT data is high perplexity.

3. This paper focuses on math and code. Do you foresee any particular challenges adapting your work to other domains?

**Ethical Concerns:**

["NO or VERY MINOR ethics concerns only"]

**Final Justification:**

The paper is well-written and focused, exploring a central intuitive idea that finetuning with high perplexity tokens (as measured by the model being finetuned) leads to more forgetting. The rebuttal by the authors included an extended learning rate sweep which now provides a more reasonable baseline for their proposed STM method. This addressed my primary concern and thus I have raised my score.

**Limitations:**

Yes, the authors have adequately discussed the limitations.

**Quality:**

3

**Strengths And Weaknesses:**

**Strengths**

The paper is well-written and focused, exploring a central intuitive idea in a well-motivated series of steps. It starts by looking at the relationship between perplexity and task degradation for datasets generated by Self-Output and Rephrase for a variety of models. It then investigates the average perplexity for Self-Output and Rephrase for Llama 3-8B Instruct and Gemma 2 IT 2B, showing that while both reduce the average PPL as compared to the ground truth, Self-Output gives the lowest average PPL at the highest computational cost. Experiments are then performed to compare SFT on these datasets, baseline finetuning, and selective token masking. The methods furthermore would be straight-forward for practitioners to incorporate into their work.

Useful ablations are performed, varying the threshold used for token masking and using instead the perplexity of a larger model in the same family for masking. Comparisons are made to regularization strategies such as weight decay and dropout. STM is shown to also improve performance for common parameter-efficient finetuning methods like LoRA and DoRA. The paper cites closely related work and explains the difference in methodology.

Overall, I think the high-level design of the experiments does a good job of thoroughly exploring the idea of using low-perplexity tokens to mitigate forgetting. My primary concern is that it is unclear whether the experiments were performed with sufficient hyperparameter sweeps to answer the experimental questions as discussed next.

**Weaknesses**

I did not find a discussion of the hyperparameters used for finetuning and in particular the learning rate. Given that the baseline finetuning performs quite poorly on target task in addition to substantial forgetting (e.g. in Table 2), I want to make sure a learning rate sweep was performed to establish a reasonable comparison. The performance of SFT is often quite sensitive to the learning rate (on both target task and in mitigating forgetting) and typically must be tuned for individual datasets. While ideally a sweep would be performed for all settings, I am most interested in the baseline given the time constraints of the paper discussion. Addressing this concern is the main improvement that would cause me to raise my score.

---

> ### Author Rebuttal · Authors · 2025-07-30
>
> We sincerely thank the reviewer for the thorough and constructive comments. Please find the response to your questions below.
>
> > lr sweeping
>
> We thank the reviewer for highlighting the importance of learning rate tuning in SFT performance and its impact on forgetting. We apologize for the lack of clarity in our original submission and will include additional details in Appendix E in the camera-ready version.
> Following prior works [9] and the Llama 2 finetuning recipe (Touvron et al.), our initial experiments used learning rates ranging from 1e-4 to 5e-5, with batch size from 4 to 16, the AdamW optimizer (bnb_8bit), a weight decay of 0.05, and LoRA rank 16.
> We also provide a weight decay analysis in Appendix I (Fig. 11).
> In response to the reviewer’s suggestion, we conducted additional learning rate sweeps down to 1e-7 for both Llama 3 8B and Gemma2 2B on the MBPP dataset. We found that while baseline SFT improves at smaller learning rates (e.g., TI: -10.6% → 1.33%), the forgetting problem (BWT) persists (e.g., -1.6%). In contrast, STM consistently outperforms baseline across all learning rates with stronger target task performance and substantially lower forgetting. Notably, STM achieves better BWT and stable performance even at small learning rates (e.g., TI: 2.23%, BWT: +1.39% at 1e-7).
> These results demonstrate that STM is more robust to learning rate choices, requires less learning rate tuning, and achieves more stable training dynamics, highlighting its practical advantage in real-world fine-tuning scenarios.
>
> Baseline Training vs STM:
>
> | Llama 3 8B-it | lr      | TI        | BWT |
> | ---|---|---|---|
> | baseline ft  | 1e-4 | -10.6%|   - |
> | baseline ft (now)     |2e-5  | -2.23%|-34.7% |
> | baseline ft  |5e-6  | 0.9%   |   - |
> | baseline ft  |1e-6  | -4.47%|   - |
> | baseline ft  |5e-7  | -1.3%  |   - |
> | baseline ft  |1e-7  | 1.33% |  -1.6% |
> |    STM        |5e-6  |  2.68%|   -0.1%|
> |    STM        |1e-6  |  3.12%|   0.53%|
> |    STM        |5e-7  |  3.12%|  1.23%|
> |    STM        |1e-7  |  2.23%|  1.39%|
> | STM (now) |2e-5   | 3.2%  |  0.2%|
>
> | Gemma 2B-it| lr      |   TI        | BWT|
> | --- | ---| ---| ---|
> |   baseline ft (now)       |1e-5  |-15.5%  | -38.2%|
> | baseline ft  |5e-6  | -4.0%   |   -|
> | baseline ft  |1e-6  | -17.6% |   -|
> | baseline ft   |1e-7  | -0.53%|  -4.7%|
> |    STM        |5e-6  |  -3.0%  | -1.1%|
> |    STM        |1e-6  | -1.5%   | -0.35%|
> |    STM        |1e-7  | 0.51%  |  0.7%|
> |    STM  (now)      |2e-5  | -0.5%   | -0.3%|
>
> > checking Rephrase with ground truth data
>
> We appreciate the concern raised by the reviewer.
> We have a sanity check on the output of Rephrase, and remove the incorrect label output as the final training set for Rephrase to ensure that all the final outputs of Rephrase are correct. But we did not evaluate the intermediate output of Rephrase, e.g., the reasoning or CoT process it could generate. This process is partially reflected in Appendix Table 13, which shows the number of training samples retained after filtering. We agree that this could have been more clearly stated and will improve the wording and methodology description in the future version to better reflect this filtering step. We appreciate the opportunity to clarify this point.
>
> > perplexity of the ground truth reponse to larger model
>
> We appreciate the reviewer’s insightful comment. As noted in our limitations, we did not explore models >10B due to resource constraints. This reflects a common real-world scenario, where most research teams work with models under 10B due to limited compute access. In practice, small LLMs (<10B) are commonly used for both training and deployment, and we believe STM remains highly relevant in such settings.
> In addition, in Appendix C, we have tested another newer 7B (OLMo) and larger Gemma 2 9B model results, and the results are consistent with our hypothesis.
> Due to time limitations, we haven’t finished the whole evaluation on the 70B model, but we will include the analysis in the future version.
>
> > focus on math and code, forsee any challenge
>
> We thank the reviewer for thoughtful feedback. We included a knowledge-based dataset as a non-target evaluation set for BWT calculation in Section 4.1. But to brief more about such limitation, the reason we focus on training code and math is that the labels are easily verifiable, and our self-output baselines are limited in such a setting for comparison. We do believe STM applies to other tasks like Knowledge QA (as ARC we mentioned in 4.1), or safety and instruction following tasks mentioned in the Limitation section. A particular challenge we foresee could be the different proportions of low perplexity tokens that exist in those tasks. And specific function words may be exceptions for filter, a much smarter masking is potentially for future work, e.g., not only on token masking but also attention to specific neurons. And based on such a setting, how do we automatically learn a good threshold for STM is also critical. To explore other domains, we also compare STM with baseline fine-tuning on the instruction following domain:
>
> Instruction following (IFEVAL). We found a similar trend for STM that sustains or even improves the non-target domain very well.
> |Gemma 2 2B-it | lr         | stm thr |  average score|
> | --- | --- | --- | ---|
> |    Original        |  -         |   -         |  0.5755|
> |    basline ft      |  1e-7   |   -         |  0.5805|
> |    basline ft      |  2e-5   |   -         |  0.6018|
> |    STM            |  1e-7   |  2.5       |  0.5970|
> |    STM            |  2e-5   |  2.5       |  0.6192|
>
> | Llama 3 8B-it   | lr         | stm thr |  average score|
> | --- | --- | --- | --- |
> |    Original        |  -         |   -         |  0.7365 |
> |    basline ft      | 1e-7    |   -         |  0.7373 |
> |    basline ft      |  2e-5   |   -         |  0.6883 |
> |     STM            | 1e-7    |  2.5      |  0.7425 |
> |    STM             | 2e-5    |  2.5      |  0.7379 |

---

> > ### Comment · Reviewer_poeA · 2025-08-06
> > **Response to Rebuttal**
> >
> > I appreciate the authors' thorough response. The results from the extended learning rate sweep match much more closely with my expectations for the baseline and illustrate my point that the performance of SFT is quite sensitive to the learning rate (on both target task and in mitigating forgetting). The updated results now give a more meaningful perspective on the difference between baseline finetuning and STM.
> >
> > In my second question concerning larger models, I did not intend to imply fine-tuning a 70B model. Rather, I meant simply computing the perplexity on the ground truth as done in Table 1 for the initial model. The idea was to determine if the average perplexity was much lower for the larger models which would imply fewer tokens would be masked out if the same threshold was used for STM.
> >
> > Overall, the authors rebuttal has addressed my main concerns and I have raised my score to recommend acceptance.

---

> > > ### Author Response · Authors · 2025-08-07
> > >
> > > Thank you very much for your updated evaluation and encouraging feedback. We sincerely appreciate your thoughtful engagement throughout the review process. We're glad that the clarifications in our rebuttal helped address your concerns, and we're encouraged by your support for the paper's acceptance.
> > >
> > > We also appreciate your suggestions regarding calculation of a larger model's avg ppl on our training dataset, and we will incorporate these improvements into the revised paper.

---

### Note · Authors · 2025-08-12

We sincerely thank the AC and all reviewers for their constructive feedback, thoughtful questions, and engagement throughout the review process. The discussions clarified key points and strengthened the work.
### Strengths highlighted by reviewers
Reviewers noted the paper’s clear writing and focus, the intuitive and well-motivated central idea, and the compelling story supported by thorough empirical analysis. They appreciated the systematic exploration of the link between high-perplexity tokens and forgetting, useful ablations (thresholds, larger-model perplexity), and comparisons with regularization strategies. STM was recognized as simple, computationally efficient, and easy to incorporate, while still competitive with far more expensive baselines like self-output. Experiments spanned multiple models, domains, and parameter-efficient tuning methods (LoRA, DoRA), with detailed cross-domain generalization results and low computational cost—important for scaling to more tasks.
### Rebuttal summary
1. **Learning rate sensitivity & robustness** – Following Reviewers poeA and TbJZ, we ran extensive LR sweeps (down to 1e-7) on Llama 3 8B-it and Gemma 2 2B-it. While baseline SFT is highly sensitive to LR—sometimes improving in-domain but still forgetting—STM consistently yields stronger target performance and lower forgetting across all rates, reducing hyperparameter tuning needs.
2. **Broader applicability** – We extended experiments to instruction following (IFEVAL) and safety (Advbench). STM sustains or improves non-target performance in IFEVAL and shows no safety degradation in Advbench, supporting real-world applicability to label-verifiable domains.
3. **Diversity & underfitting** – Addressing the concerns raised by Reviewers PAG5, TbJZ, and C8Fr, diversity analyses (self-BLEU on math reasoning) show STM matches SFT diversity while reducing forgetting. Even when masking symbolic tokens, STM improves both target and non-target performance, suggesting sufficient retained context. We note curriculum learning and conditional masking as promising refinements.
4. **Theoretical perspective** – While primarily empirical, LoRA rank and weight-update analyses show STM induces smaller, more localized updates, potentially preserving general capabilities. A formal theory is important, but consistent cross-model, cross-domain trends offer a solid empirical foundation.

---

### Decision · Program_Chairs · 2025-09-17

**Decision:**

Accept (poster)

**Comment:**

**Summary**

This paper addresses catastrophic forgetting in large language model fine-tuning by proposing Selective Token Masking (STM), a method that masks high-perplexity tokens during supervised fine-tuning. The central empirical finding is that fine-tuning on high-perplexity tokens correlates with increased forgetting of general capabilities, while low-perplexity tokens enable better preservation of non-target performance.

**Reason to Accept**
-  The paper provides a compelling and intuitive explanation for why generated data helps mitigate forgetting—the connection between token perplexity and catastrophic forgetting represents a valuable empirical discovery that could influence future fine-tuning practices.

- STM offers a practical, low-cost alternative to expensive methods like Self-Output generation, making it accessible for resource-constrained settings while maintaining competitive performance.

**Summarize the discussion**

- Reviewer **poeA** initially raised concerns about hyperparameter selection and baseline performance but was convinced by the authors' extensive learning rate sweeps. The additional experiments showed STM's robustness across learning rates and confirmed its practical advantages. This reviewer explicitly raised their score to Accept.

- Reviewer **PAG5** appreciated the additional experiments on diversity analysis and instruction-following tasks but maintained concerns about limited domain coverage and potential long-term effects of low-perplexity training. Despite acknowledging the quality of responses, they maintained their Borderline Accept rating.

- Reviewer **TbJZ** remained unconvinced by explanations for baseline performance degradation, viewing it as indicative of fundamental experimental flaws. While the authors provided additional results demonstrating improved baseline performance through proper hyperparameter tuning, **this reviewer did not engage in subsequent discussions or respond to the Area Chair's invitation for further dialogue**. Based on the comprehensive additional experimental evidence provided in the rebuttal—including extensive learning rate sweeps that showed positive target task improvements (+1.33% on MBPP) and STM's consistent superiority across all tested learning rates—the Area Chair determined that the core experimental validity concerns had been adequately addressed.

- Reviewer **C8Fr** acknowledged the empirical evidence but expressed concerns about the lack of theoretical justification. However, this critique may be misplaced given that the authors never claimed to provide theoretical foundations, and the current state of LLM research lacks comprehensive theoretical frameworks for understanding model behavior. **In such a context, demanding rigorous theoretical justification for empirical discoveries may set an unrealistic standard**. In the rapidly evolving LLM landscape, **empirical discoveries that consistently reproduce across diverse experimental settings provide substantial value to the research community**. The insight connecting token perplexity to catastrophic forgetting represents a meaningful contribution to our empirical understanding of fine-tuning dynamics and offers immediately actionable insights for practitioners.

**Overall**

The authors demonstrated strong engagement, providing extensive additional experiments and thoughtful responses. The fundamental theoretical concerns raised by reviewers were not fully resolved, leading to mixed opinions. However, the empirical rigor, practical utility, and reproducible findings outweigh the theoretical limitations in the current context of LLM research